# Language Model Uncertainty Quantification with Attention Chain

**Yinghao Li, Rushi Qiang, Lama Moukheiber, Chao Zhang**
Georgia Institute of Technology, Atlanta, USA
{yinghaoli,rqiang6,lmoukheiber3,chaozhang}@gatech.edu

## Abstract

Accurately quantifying a large language model's (LLM's) predictive uncertainty is crucial for judging the reliability of its answers. Although current uncertainty quantification (UQ) research still concentrates on short, direct answers, typically in closed-form formats (*e.g.*, multiple choice), intermediate reasoning steps are becoming increasingly critical to LLM responses. It complicates UQ because the probabilities assigned to answer tokens are conditioned on a vast space of preceding reasoning tokens. Direct marginalization is infeasible, and this dependency inflates probability estimates, causing overconfidence in UQ. To address this, we propose UQAC, an efficient method that narrows the reasoning space to a tractable size for marginalization. UQAC iteratively constructs an "attention chain" of tokens deemed "semantically crucial" to the final answer via a back-tracking procedure. Starting from the answer tokens, it uses attention weights to identify the most influential predecessors, then iterates this process until reaching the input tokens. The resulting chain is further refined with similarity filtering and probability thresholding, which together reduce the reasoning space, facilitating the approximation of marginal answer-token probabilities. We validate UQAC on multiple reasoning benchmarks with advanced open-source LLMs, demonstrating that it consistently delivers reliable UQ estimates with high computational efficiency.

## 1 Introduction

As large language models (LLMs) become integral across domains, ensuring their reliability is increasingly important. This issue is especially pressing in high-stakes scenarios such as medical diagnosis (Fox, 1980; Simpkin & Schwartzstein, 2016; Shen et al., 2023), where incorrect predictions can have severe consequences. In such domains, uncertainty quantification (UQ) is crucial for estimating a model's confidence in its answer, detecting when it operates beyond its knowledge scope, and mitigating phenomena like hallucinations (Li et al., 2023; 2024b; Xu et al., 2024). However, efficient methods for UQ in natural language generation remain scarce (Gawlikowski et al., 2023; Kuhn et al., 2023), underscoring the need for robust techniques to assess LLM reliability.

Unlike auto-encoding models (Peters et al., 2018; Devlin et al., 2019) that produce outputs within constrained spaces, autoregressive LLMs (Radford et al., 2018), particularly those prompted or fine-tuned to generate detailed reasoning steps (Wei et al., 2022; OpenAI, 2024; DeepSeek-AI, 2025), yield arbitrarily long reasoning sequences. These sequences can be seen as samples from an intractably large and high-dimensional language space, making direct marginalization of intermediate steps impractical. Existing approaches often rely on approximating this space with a tractable semantic representation (Kuhn et al., 2023; Hou et al., 2024; Jiang et al., 2023; Ling et al., 2024; Lin et al., 2024b; Chen & Mueller, 2024) But they frequently employ Self-Consistency sampling (Lakshminarayanan et al., 2017; Wang et al., 2023) to repeatedly generate answers using the same or paraphrased prompts, which can be computationally expensive. In another line, some methods focus on isolating semantically critical tokens (Duan et al., 2024; Lin et al., 2024a), yet they are typically tailored to short outputs and struggle with extended reasoning sequences. Furthermore, approaches

for explicit verbalization of confidence (Lin et al., 2022; Tian et al., 2023; Xiong et al., 2024) demand additional model training, creating extra barriers for end users.

To overcome these challenges, we introduce a model-agnostic UQ method, *Uncertainty Quantification with Attention Chain* (UQAC), designed to accommodate intermediate reasoning sequences without extra fine-tuning. UQAC builds on the observation that *not all tokens in a reasoning sequence equally contribute to the final answer*—a concept aligned with "attention" mechanisms (Bahdanau et al., 2015; Luong et al., 2015; Vaswani et al., 2017) and prior findings (Duan et al., 2024; Lin et al., 2024a). Specifically, UQAC traces "semantically crucial" tokens by iteratively backtracking through the reasoning sequence: starting from the final answer tokens and using attention weights to identify their most influential predecessors (Figure 1). This process continues until reaching the input tokens, forming an *attention chain* that captures the reasoning steps most critical to the final outcome (§ 4.1). To further refine the attention chain, UQAC prunes peripheral tokens by assessing their similarity to the answer tokens (§ 4.2), thus filtering out syntactic or auxiliary elements. We then apply probability thresholding to remove token combinations with minimal influence on the final answer's probability distribution, producing a tractably small set of key reasoning paths (§ 4.3). This pared-down set is used to compute marginal probabilities of the final answer, effectively measuring the model's confidence. Compared with existing methods, UQAC provides confidence scores bounded between 0 and 1 in a single pass, yielding more interpretable results and improved calibration. Overall, UQAC offers:

- **Applicability**: Works with any white-box autoregressive language models without the need for model or hyper-parameter tuning.
- **Scalability**: Handles responses of arbitrary length by isolating the most influential tokens.
- **Efficiency**: Runs in parallel with autoregressive generation and requires no additional sampling or external model inference.
- **Calibration**: Provides bounded scores that directly reflect the model's confidence.

Our experiments across diverse LLMs and datasets (§ 5) demonstrate that UQAC delivers superior calibration performance among UQ methods with comparable overhead (§ 6). Code and data are available at `https://github.com/Yinghao-Li/UQAC`.

## 2 Related Works

Early approaches to UQ in autoregressive models compute cumulative token-wise probabilities or entropies (Kadavath et al., 2022; Kuhn et al., 2023), as well as their length-normalized variants (Malinin & Gales, 2021). However, these methods struggle with long reasoning sequences, where semantic and auxiliary tokens are conflated in joint probability estimations. To address the intractable reasoning space, several works approximate it through a semantic space composed of model outputs sampled from multiple runs (Kuhn et al., 2023; Hou et al., 2024; Jiang et al., 2023; Ling et al., 2024; Lin et al., 2024b; Chen & Mueller, 2024). For instance, Kuhn et al. (2023) propose semantic entropy, which clusters similar responses and measures uncertainty via normalized cumulative log probabilities, later extended by Lin et al. (2024b) to black-box models using a co-occurrence entailment matrix. Chen & Mueller (2024) combine semantic entropy with self-reflective evaluation for enhanced robustness. Similarly, Jiang et al. (2023) employ paraphrased prompts to gather multiple outputs, while Hou et al. (2024) differentiate aleatoric and epistemic uncertainty by appending various clarification sequences to the original query. All these methods require multiple model runs, mirroring Self-Consistency (Wang et al., 2023), which is expensive due to the autoregressive structure of decoder-based LLMs. In addition, approaches like semantic entropy rely on external encoder models for clustering, further increasing computational overhead.

A second line of research, akin to UQAC, aims to isolate auxiliary tokens and evaluate token-level contributions to the final answer semantics (Duan et al., 2024; Lin et al., 2024a). Duan et al. (2024) propose SAR, which quantifies token importance as the negative similarity between the complete sequence and a version with the token removed, weighting token-level entropies accordingly. Although effective, SAR necessitates multiple forward passes for thorough sequence-level aggregation. Lin et al. (2024a) introduce CSL, which modifies sequence probabilities by weighting tokens based on attention scores captured during the

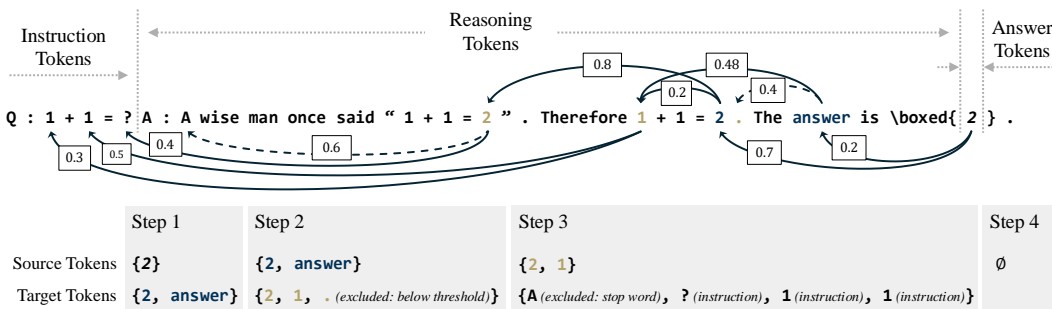

Figure 1: Construction of the attention chain $x_{\text{attn}}$ with a 3-step attention backtracking procedure. Arrows point from source tokens to target tokens with the top-2 attention weights (displayed on the arrows), *i.e.*, $L_{\text{tgt}} = 2$. Solid lines indicate valid target tokens, while dashed lines indicate invalid ones for various reasons explained below. For simplicity, we do not shift the source set $x_{\text{src}}$ as described in appendix B.2.

model's prediction of self-verification tokens (*e.g.*, "Y" or "N"). While CSL requires only a single pass, it depends on a large in-domain validation set (*e.g.*, 1,000 samples) to select attention heads, which is impractical in most real-world contexts. Moreover, both SAR and CSL are limited to short responses, as SAR's complexity grows factorially and CSL's localized attention scores fail to capture long-range dependencies.

A third line of work explores prompting LLMs to express uncertainty in natural language (Lin et al., 2022; Tian et al., 2023; Xiong et al., 2024; Amayuelas et al., 2024). Lin et al. (2022) demonstrate that GPT-3 can be fine-tuned to output well-calibrated Verbalized Uncertainty without relying on logits. Tian et al. (2023) show that confidence statements produced by RLHF-trained models can outperform traditional probability-based approaches, even under distribution shifts. Meanwhile, Xiong et al. (2024) propose a black-box framework featuring tailored prompts, sampling, and aggregation to reduce overconfidence, and Amayuelas et al. (2024) improve calibration through targeted fine-tuning on specialized datasets. Nonetheless, they typically require additional training or domain-specific adaptations, limiting their applicability to a broader range of models and tasks.

## 3 Problem Definition

Consider an instruction sequence $x_{\text{instr}} \in \mathbb{V}^{L_{\text{instr}}} \triangleq [x_1, x_2, \ldots, x_{L_{\text{instr}}}]$ of length $L_{\text{instr}}$, where $\mathbb{V}$ is a vocabulary set. A language model $\mathcal{M}$ generates a response sequence $x_{\text{resp}} \in \mathbb{V}^{L_{\text{resp}}}$ of length $L_{\text{resp}}$, which can be split into a Chain-of-Thought (CoT; Wei et al., 2022) reasoning sequence $x_{\text{cot}} \in \mathbb{V}^{L_{\text{cot}}}$ and the final answer $x_{\text{ans}} \in \mathbb{V}^{L_{\text{ans}}}$. We write $x_{\text{resp}} = x_{\text{cot}} \oplus x_{\text{ans}}$ (Figure 1), where $\oplus$ represents "concatenation", disregarding any tokens after $x_{\text{ans}}$.

Our objective in UQ is to determine the model's confidence in its final answer, denoted by $P_{\mathcal{M}}(x_{\text{ans}}|x_{\text{instr}})$. Directly using the joint probability $P_{\mathcal{M}}(x_{\text{resp}}|x_{\text{instr}}) = P_{\mathcal{M}}(x_{\text{ans}}, x_{\text{cot}}|x_{\text{instr}})$ is not suitable, as the reasoning sequence $x_{\text{cot}}$ typically includes both semantically crucial tokens needed for deriving the answer and auxiliary tokens that primarily serve syntactic or coherence purposes. These auxiliary tokens often make up the bulk of $x_{\text{cot}}$ yet have minimal impact on the actual answer semantics. Consequently, using $P_{\mathcal{M}}(x_{\text{resp}}|x_{\text{instr}})$ as a confidence measure can be both biased and unintuitive: the probability tends to decrease monotonically with the length of the response, leading to small joint probabilities that are difficult to interpret. Moreover, conditioning the final answer's confidence on $x_{\text{cot}}$ is unnecessary when the primary concern is the correctness and trustworthiness of the final answer itself. Therefore, this confidence formally is expressed as

$$P_{\mathcal{M}}(x_{\text{ans}}|x_{\text{instr}}) = \sum_{x_{\text{cot}}} P_{\mathcal{M}}(x_{\text{ans}}|x_{\text{cot}}, x_{\text{instr}}) P_{\mathcal{M}}(x_{\text{cot}}|x_{\text{instr}}), \qquad (1)$$

where $x_{\text{cot}}$ ranges over all possible reasoning sequences in $\mathbb{V}^{L_{\text{cot}}}$. However, the summation is computationally intractable: even when the vocabulary dimension is on the order of

10,000, the space grows exponentially with the sequence length $L_{\text{cot}}$, which can reach into the hundreds. Accordingly, the main challenge in LLM UQ is to construct an accurate approximation $\widetilde{P}_{\mathcal{M}} \approx P_{\mathcal{M}}(x_{\text{ans}}|x_{\text{instr}})$ that faithfully reflects the model's confidence, without explicitly summing over all possible reasoning sequences.

In the following, we focus on a single instance of a single model and thus omit the model indicator $\mathcal{M}$ and the instance index. Table 4 summarizes the notations used throughout.

## 4 Uncertainty Quantification with Attention Chain

### 4.1 Attention Chain

We first apply *attention backtracking* to identify semantically crucial tokens $x_{\text{attn}} \sqsubset x_{\text{cot}}$, , where $\sqsubset$ is defined as a "proper (not necessarily consecutive) subsequence", *i.e.*, $x_1 \in \mathbb{V}^N \sqsubset x_2 \in \mathbb{V}^M \leftrightarrow \exists 1 \leqslant i_1 < \cdots < i_N \leqslant M \text{ s.t. } x_{1,[k]} = x_{2,[i_k]}, \forall k \in \mathbb{N}_{[1,N]}$. Attention backtracking itself is an iterative procedure governed by a function $f: x_{\text{src}} \mapsto x_{\text{tgt}}$, which selects a target subsequence $x_{\text{tgt}} \sqsubset x_{\text{cot}}$ deemed most influential for predicting the source tokens $x_{\text{src}} \sqsubset x_{\text{resp}}$, based on the model's attention weights. We impose a buffer size $L_{\text{tgt}} \triangleq \max |x_{\text{tgt}}|$ that bounds the number of tokens selected at each step. The process starts with $x_{\text{src}}^{(0)} = x_{\text{ans}}$ and $x_{\text{attn}}^{(0)} = \varnothing$. At each iteration $z$, we compute $x_{\text{tgt}}^{(z)} = f(x_{\text{src}}^{(z-1)})$ and update $x_{\text{src}}^{(z)} = x_{\text{tgt}}^{(z)}$. This continues until no new tokens are extracted from $x_{\text{cot}}$. The final sequence $x_{\text{attn}}$ is then the concatenation of all target sequences $x_{\text{attn}} \triangleq x_{\text{tgt}}^{(0)} \oplus \cdots \oplus x_{\text{tgt}}^{(Z)}$, assuming $f$ is applied $Z$ times. $x_{\text{attn}}$ serves as a compact semantic approximation of the original reasoning chain $x_{\text{cot}}$. Figure 1 provides an illustration of attention backtracking. Each step of $f$ consists of three stages: 1) attention weight processing; 2) attention head selection and aggregation; and 3) target token identification, detailed in the following paragraphs.

**Attention Weight Processing** For a sequence of length $T$, LLMs employ self-attention (Vaswani et al., 2017) to compute a weight for each token in the prefix $x_{\leqslant T}$ (which includes the instruction, *i.e.*, $x_{\text{instr}} \sqsubset x_{\leqslant T}$) to gauge its contribution to predicting the next token $x_{T+1}$. In an $L$-layer, $H$-head Transformer model $\mathcal{M}$, the attention weight vector for token $x_T$ in the $l$-th layer and $h$-th head is defined as

$$\boldsymbol{\alpha}_T^{(l,h)} \in [0,1]^T = \text{softmax}(K_T^{(l,h)} q_T^{(l,h)} / \sqrt{d_k}), \tag{2}$$

where $q_T^{(l,h)} \in \mathbb{R}^{d_k}$ is the query vector for $x_T$, $K_T^{(l,h)} \in \mathbb{R}^{T \times d_k}$ is the key matrix for the prefix $x_{\leqslant T}$, and $d_k$ is the dimensionality of the key vectors. This attention vector determines how much semantic information from $x_{\leqslant T}$ is transferred to $x_{T+1}$, thereby serving as an indicator of the relative importance of each token in the prefix.

However, LLMs often overemphasize tokens that are close to position $T$, irrespective of their semantic relevance. To mitigate this bias, we reweight the most recent $C$ attention weights by multiplying them with a sequence of decreasing factors $\gamma \in [0,1]^C$. Moreover, since some LLMs tend to assign disproportionately high attention to the instruction's BOS token, we reset that particular attention weight to zero. After these modifications, the attention vector is renormalized as follows (with layer and head indices omitted for brevity):

$$\boldsymbol{\alpha}_{T,[T-C:T]} \leftarrow \gamma \odot \boldsymbol{\alpha}_{T,[T-C:T]}; \quad \alpha_{T,[1]} \leftarrow 0; \qquad \boldsymbol{\alpha}' = \boldsymbol{\alpha}/\textstyle\sum \boldsymbol{\alpha}, \tag{3}$$

where "$\odot$" denotes element-wise multiplication and the square brackets indicate a slice. Please refer to appendix B.1 for further discussion on the reweighting factors $\gamma$.

**Attention Head Selection and Aggregation** Different attention heads typically capture distinct aspects of the input sequence. Not all attention heads contribute equally when predicting $x_{T+1}$, making it crucial to identify the most informative ones for discerning semantically significant tokens (Lin et al., 2024a). To enable a *lightweight*, *training-free* selection process, we leverage attention entropy, defined as $\mathcal{H}(\boldsymbol{\alpha}) = -\sum \boldsymbol{\alpha} \log \boldsymbol{\alpha}$, which

quantifies the uncertainty of the attention distribution. A higher entropy value indicates a more uniform distribution, suggesting a less informative attention head. Consequently, we select the top-$K$ heads with the lowest entropy:

$$(l, h)_{T,k} = \underset{(l,h)}{\arg\min}_k \mathcal{H}(\boldsymbol{\alpha}_T'^{(l,h)}); \quad k \in \mathbb{N}_{[1,K]}, \tag{4}$$

where the operator $\arg\min_k$ returns the *index* corresponding to the $k$-th smallest entropy value, and the subscript $T$ emphasizes that the computation is performed independently for each token $x_T$. Subsequently, we aggregate the selected attention weights into a single vector $\boldsymbol{\alpha}_T^* \in [0,1]^T$ via element-wise maximization:

$$\alpha_{T,[t]}^* = \max_k \alpha_{T,[t]}'^{(l,h)_{T,k}}; \quad t \in \mathbb{N}_{[1,T]}. \tag{5}$$

Due to the maximization, $\sum \boldsymbol{\alpha}_T^* \in (1, T]$, *i.e.*, $\boldsymbol{\alpha}_T^*$ no longer resides on a probability simplex.

**Target Token Identification**    Let $t_s \in \mathbb{N}_{[1,T]}$ for $s \in \mathbb{N}_{[1,L_{\mathrm{src}}]}$ denote the positions of source tokens within the source token set $\boldsymbol{x}_{\mathrm{src}} \triangleq \{x_{t_s}\}_s$ (with step index $\cdot^{(z)}$ omitted). Here we treat $\boldsymbol{x}_{\mathrm{src}}$ and $\boldsymbol{x}_{\mathrm{tgt}}$ as *sets* for convenience. Given the set of attention weights $\{\boldsymbol{\alpha}_{t_s}^*\}_s$ corresponding to $\boldsymbol{x}_{\mathrm{src}}$, our goal is to construct $\boldsymbol{x}_{\mathrm{tgt}}$ by selecting the top-$L_{\mathrm{tgt}}$ tokens based on their cumulative attention weights. For each token position $t \in \mathbb{N}_{[1,T]}$, we compute a cumulative attention weight and gather the target set as

$$\phi_{[t]} = \sum_{s=1}^{L_{\mathrm{src}}} \alpha_{t_s,[t]}^*; \quad \boldsymbol{x}_{\mathrm{tgt}} = \{x_t \mid \phi_{[t]} > \theta; \; \mathbb{I}(x_t \notin \mathbb{V}_{\mathrm{stop}}); \; t \in \{\underset{t}{\arg\max}_k \phi_t\}_{k=1}^{L_{\mathrm{tgt}}}\}. \tag{6}$$

where $\theta$ is a threshold to filter out weakly attended tokens, and the operator $\arg\max_k$ returns the indices corresponding to the $k$-th largest cumulative weight.

Finally, we define the step-wise backtracking function $f$ as the sequential composition of (2)–(6). The attention chain $\boldsymbol{x}_{\mathrm{attn}}$ is generated by iteratively applying $f$; at each step $z$, a new target set is produced, and the final attention chain is the concatenation of all these target sets, as discussed above. For further details, please refer to appendix B.2.

## 4.2   Similarity-Based Filtering

Since the backtracking function $f$ does not explicitly regulate the attention chain length $L_{\mathrm{attn}}$, the chain may be long, complicating the formation of a tractable reasoning space. Therefore, we employ a similarity-based filtering strategy for finer control. Given the answer tokens $\boldsymbol{x}_{\mathrm{ans}}$ and the attention chain $\boldsymbol{x}_{\mathrm{attn}}$, we compute the similarity between each token pair $(x_m, x_n)$, where $x_m \sqsubset \boldsymbol{x}_{\mathrm{ans}}$ and $x_n \sqsubset \boldsymbol{x}_{\mathrm{attn}}$, by applying cosine similarity to their last-layer output logits $\boldsymbol{h}_m$ and $\boldsymbol{h}_n$. We calculate the similarity vector $\boldsymbol{w} \in [-1,1]^{L_{\mathrm{attn}}}$ as:

$$\mathrm{sim}(x_m, x_n) = \boldsymbol{h}_m^\mathsf{T}\boldsymbol{h}_n / \|\boldsymbol{h}_m\|\|\boldsymbol{h}_n\|, \quad w_{[n]} = \sum_m \mathrm{sim}(x_m, x_n), \quad m \in \mathbb{N}_{[1,L_{\mathrm{ans}}]}, \; n \in \mathbb{N}_{[1,L_{\mathrm{attn}}]}. \tag{7}$$

We retain the tokens with the top-$L_{\mathrm{attn}}'$ weights to form the filtered sequence $\boldsymbol{x}_{\mathrm{attn}}' \sqsubseteq \boldsymbol{x}_{\mathrm{attn}}$:

$$\boldsymbol{x}_{\mathrm{attn}}' = [x_{\mathrm{attn},[n_1]}, \ldots, x_{\mathrm{attn},[n_i]}, \ldots] \; s.t. \; w_{[n_i]} > 0; \; n_i \in \{\underset{k}{\arg\max}_k \boldsymbol{w}\}_{k=1}^{L_{\mathrm{attn}}'}, \tag{8}$$

where the threshold of 0 excludes tokens with negative similarity scores. Please refer to appendix B.3 for further discussion.

## 4.3   Model Confidence

Using the attention chain $\boldsymbol{x}_{\mathrm{attn}}$ and its filtered version $\boldsymbol{x}_{\mathrm{attn}}'$, we first define two confidence approximations in (1) based on the joint probabilities of the attention chain and answer tokens: **1) attention approximation** $\widetilde{P}_{\mathcal{M},\mathrm{attn}} \triangleq P(\boldsymbol{x}_{\mathrm{ans}}, \boldsymbol{x}_{\mathrm{attn}} | \boldsymbol{x}_{\mathrm{instr}})$; **2) similarity approximation** $\widetilde{P}_{\mathcal{M},\mathrm{sim}} \triangleq P(\boldsymbol{x}_{\mathrm{ans}}, \boldsymbol{x}_{\mathrm{attn}}' | \boldsymbol{x}_{\mathrm{instr}})$. Since $\boldsymbol{x}_{\mathrm{attn}}' \sqsubseteq \boldsymbol{x}_{\mathrm{attn}}$, we have $\widetilde{P}_{\mathcal{M},\mathrm{sim}} \geq \widetilde{P}_{\mathcal{M},\mathrm{attn}}$. Both

approximations can be viewed as indicators of the model's confidence in the critical semantics captured by the final answer $x_{ans}$. By marginalizing out the filtered attention chain $x'_{attn}$ in $\widetilde{P}_{\mathcal{M},sim}$, we obtain **3) answer approximation** $\widetilde{P}_{\mathcal{M}} \triangleq P(x_{ans}|x_{instr})$, which more directly captures the model's confidence in $x_{ans}$ alone. The challenge is to reduce the approximated reasoning space $\mathbb{V}^{L'_{attn}}$ to a manageable subset $\mathbb{S}$ so that the summarization in

$$\widetilde{P}_{\mathcal{M}} = \sum_{x'_{attn} \sim \mathbb{S}} P(x_{ans}, x'_{attn}|x_{instr}) \tag{9}$$

remains computationally feasible. Approximating $x_{cot}$ by $x'_{attn}$ significantly reduces the token count, and we constrain $L'_{attn} \leq 10$. Furthermore, language models typically assign very high probabilities (*e.g.*, $> 0.99$) to the selected tokens; hence, substituting a chosen token $x'_{attn,[i]}$ with any other candidate $x'_{attn,[i]} \neq x'_{attn,[i]}$ often results in a negligible joint probability, *i.e.*, $P\left(x_{ans}, x'_{attn,[\backslash i]}, x'_{attn,[i]} \neq x'_{attn,[i]}|x_{instr}\right) \approx 0$. Consequently, we only consider alternative tokens that exceed a 0.01 probability threshold, and we alter just one token at a time while keeping the others unchanged. In our experiments, this yields on average only six such candidates per position (excluding $x'_{attn}$), so $\mathbb{E}[|\mathbb{S}|] = 7$. Although multiple forward passes are still required, all elements in $\mathbb{S}$ are independent and can be evaluated without any recurrent or sequential dependencies, in contrast to Self-Consistency approaches.

## 5 Experiment Setup

**Datasets and Models**   We use three standard reasoning datasets: 1) GSM8k (Cobbe et al., 2021) for basic math; 2) MATH (Hendrycks et al., 2021) for advanced math; and 3) BBH (bench authors, 2023; Suzgun et al., 2023) for logical, mathematical, and commonsense reasoning. See appendix C.1 for details. We evaluate nine instruction-tuned white-box SOTA LLMs spanning 1B–9B parameters: Llama-3.2-[1B,3B], Llama-3.1-8B (Llama Team, 2024), gemma-2-[2b,9b] (Gemma Team, 2024), Qwen2.5-[1.5B,3B,7B] (Qwen Team, 2024), and DeepSeek-R1-Distill-Llama-8B (DeepSeek-AI, 2025).

**Baselines**   We compare three variants of UQAC, $\widetilde{P}_{\mathcal{M},attn}$, $\widetilde{P}_{\mathcal{M},sim}$, and $\widetilde{P}_{\mathcal{M}}$, against the following baselines. 1) $P_{\mathcal{M}}(x_{ans}|x_{cot}, x_{instr})$, the joint conditional probability of the answer; 2) $P_{\mathcal{M}}(x_{resp}|x_{instr})$, the joint conditional probability of the entire response; 3) $\overline{P}_{\mathcal{M}}(x_{ans})$, the mean probability of answer tokens; 4) $\overline{P}_{\mathcal{M}}(x_{resp})$, the mean probability of response tokens; 5) **Predictive Entropy** $\mathcal{H} = -\sum_{t=L_{instr}+1}^{L_{instr}+L_{resp}} \sum_{x_t} P_{\mathcal{M}}(x_t|x_{<t}) \log P_{\mathcal{M}}(x_t|x_{<t})$ (Kuhn et al., 2023); 6) **Length-Normalized Predictive Entropy** $\overline{\mathcal{H}} = \mathcal{H}/L_{resp}$ (Malinin & Gales, 2021); 7) **Self-Consistency** (Wang et al., 2023), the probability averaged over 5 independently sampled answers; and 8) **Verbalized Uncertainty** (Xiong et al., 2024), a model-generated confidence score via additional prompts. See appendix C.2 for further discussion.

**Evaluation Metrics**   Prior works produce unbounded confidence scores (Kuhn et al., 2023; Duan et al., 2024) and thus rely on AUROC for its scale-invariance. However, while AUROC effectively captures the model's ability to distinguish correct from incorrect predictions, it provides no indication of how well the predicted confidence scores align with actual accuracy, that is, it does not reflect calibration. Consequently, when queries originate from distributions different from those seen during training or from previously tested scenarios, relying solely on AUROC yields limited insight into the true reliability of the model's answers. Therefore, we emphasize Expected Calibration Error (ECE; Naeini et al., 2015) and calibration plots, while reporting AUROC for completeness. See appendix C.3 for details.

**Implementation**   We set $C = 10$ in (3), $K = 16$ in (4), $L_{tgt} = 3$ and $\theta = 0.5$ in (6), and delay applying $\theta$ until $L_{attn} \geqslant 5$ to avoid premature backtracking. We use $L'_{attn} = 10$ in (8). For token prediction, The inference temperature is set as 0 and top-$p$ as 1. We balance the numbers of correct and incorrect predictions through subsampling. All experiments run on an NVIDIA A100-SXM4-80GB GPU with bf16 precision. Results, reported as mean $\pm$ standard deviation over five runs (seeds 0–4), are from our own implementations.

| | GSM8k | | MATH | | BBH | |
|---|---|---|---|---|---|---|
| | AUROC↑ | ECE↓ | AUROC↑ | ECE↓ | AUROC↑ | ECE↓ |
| $\overline{P}_{\mathcal{M}}(x_{\text{ans}})$ | 60.9±0.6 | 49.4±0.0 | 74.2±1.2 | 48.1±0.2 | 65.0±0.9 | 44.6±0.3 |
| $\overline{P}_{\mathcal{M}}(x_{\text{resp}})$ | 61.1±0.8 | 41.9±0.0 | 69.0±1.5 | 41.5±0.1 | 59.3±1.3 | 42.6±0.2 |
| $P_{\mathcal{M}}(x_{\text{ans}}\vert x_{\text{cot}}, x_{\text{instr}})$ | 60.7±0.6 | 48.4±0.1 | 73.9±1.2 | 43.7±0.3 | 66.1±0.8 | 38.1±0.5 |
| $P_{\mathcal{M}}(x_{\text{resp}}\vert x_{\text{instr}})$ | 66.7±0.7 | 50.0±0.0 | 79.0±1.1 | 50.0±0.0 | 63.6±1.5 | 45.3±0.4 |
| $1 - \overline{\mathcal{H}}^{(a)}$ | 61.8±0.8 | 33.2±0.1 | 69.3±1.5 | 35.3±0.2 | 59.4±1.3 | 33.5±0.4 |
| $1 - \mathcal{H}^{(a)}$ | 67.1±0.8 | - | 78.6±1.1 | - | 63.6±1.2 | - |
| Self-Consistency | 66.4±1.9 | 28.9±0.8 | 79.5±1.0 | 15.8±0.8 | 79.5±1.0 | 31.6±0.7 |
| Verbalized | 54.9±0.5 | 42.9±0.2 | 57.4±0.7 | 45.1±0.2 | 58.2±1.2 | 39.7±0.3 |
| UQAC-$\widetilde{P}_{\mathcal{M},\text{attn}}$ | 58.4±0.8 | 37.4±0.6 | 68.6±1.2 | 42.8±0.5 | 64.6±1.3 | 23.4±1.0 |
| UQAC-$\widetilde{P}_{\mathcal{M},\text{sim}}$ | 60.7±1.0 | 28.0±0.5 | 68.0±1.3 | 21.6±1.0 | 65.1±1.2 | 22.1±1.0 |
| UQAC-$\widetilde{P}_{\mathcal{M}}$ | 61.3±0.9 | 33.6±0.4 | 69.5±1.2 | 25.8±0.9 | 66.7±1.2 | 24.2±0.9 |

[a] We use $1 - \mathcal{H}$ and $1 - \overline{\mathcal{H}}$ as higher entropy indicates higher uncertainty, different from probabilities. ECE is not applicable to predictive entropy $\mathcal{H}$ as its value is unbounded.

Table 1: Performance of different UQ methods. The scores (in %) are reported as $\mu \pm \sigma$ over 5 runs, averaged across all LLMs. Higher AUROC is better; ECE the opposite.

## 6 Results

**Main Results** Table 1 summarizes the average performance across all LLMs, with additional details in appendix E, showing superior calibration from UQAC. Notably, the unnormalized baselines, *i.e.*, $P_{\mathcal{M}}(x_{\text{resp}}\vert x_{\text{instr}})$ and $\mathcal{H}$, achieve higher AUROC on GSM8k and MATH by leveraging pronounced differences in response lengths between correct and incorrect answers (shown in Figure 4a and discussed in appendix D.2). Since longer sequences yield lower joint probabilities and higher cumulative entropy, this effect benefits the AUROC calculation. However, when response lengths are more balanced, as with DeepSeek-R1 on GSM8k (Figure 4c and Table 18) or across most models on the BBH dataset due to in-context regularization, the advantage diminishes. Furthermore, $\mathcal{H}$ cannot be calibrated, and $P_{\mathcal{M}}(x_{\text{resp}}\vert x_{\text{instr}})$ suffers from a 50.0% ECE, indicating that predicted probabilities are pushed toward extremes (1 as in Figure 3a or 0 as in Figure 3b). This concentration harms the ability to distinguish correct from incorrect predictions based solely on confidence, yielding poor calibration. While the length-normalized predictive entropy $\overline{\mathcal{H}}$ performs comparably to UQAC on GSM8k, it falters with longer inputs and extended reasoning chains, as evidenced by its inferior performance on the MATH and BBH datasets and a more centralized entropy distribution (Figure 3c). In general, UQAC provides descent AUROC scores as well as delivers significantly lower ECE across datasets, aligning more closely with true predictive performance and offering more reliable uncertainty estimates (Figure 3).

**External Baselines** Without additional fine-tuning, Verbalized Uncertainty exhibits significantly inferior performance, consistently underperforming even the $\overline{\mathcal{H}}$ baseline. This suggests that it is not effectively integrated into prevalent LLMs and requires extensive training or careful prompt engineering for practical utility. In contrast, Self-Consistency, a variant of Deep Ensembles (Lakshminarayanan et al., 2017), generally achieves superior performance, albeit with significant computational overhead (shown in Figure 2 and discussed in appendix D.3), aligning with theoretical insights and prior empirical evidence (Fort et al., 2019; Wang et al., 2023; Li et al., 2024a). When efficiency is not a primary concern, it is possible to combine Self-Consistency with UQAC to further enhances calibration performance.

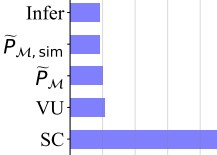

Figure 2: Run time.

**UQAC Variants** Among the variants of UQAC, $\widetilde{P}_{\mathcal{M},\text{attn}}$ yields lower AUROC and higher ECE. As reasoning becomes more complex and the attention chain $L_{\text{attn}}$ lengthens, the

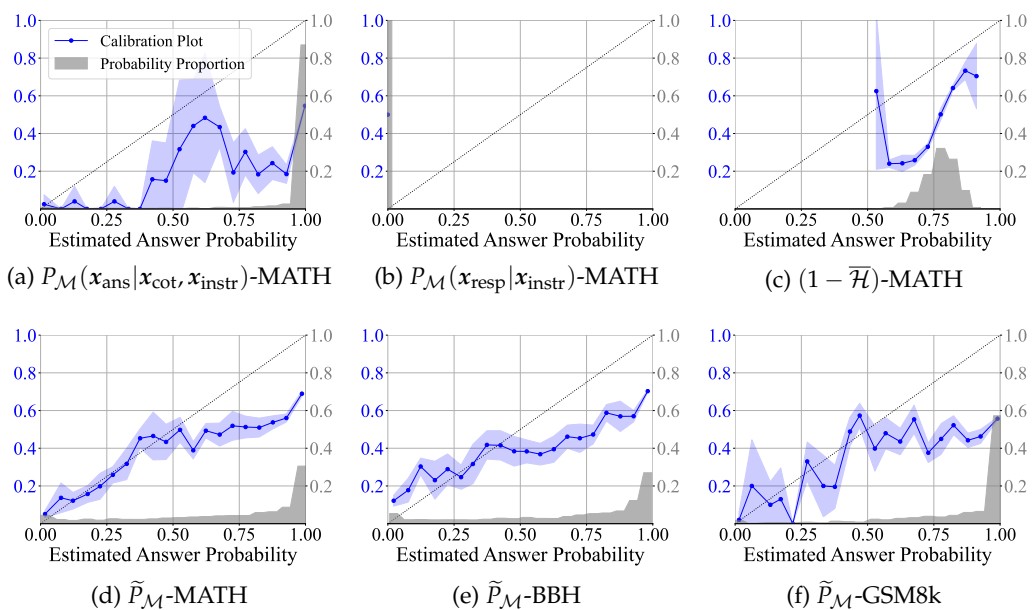

Figure 3: Calibration plots and probability histogram for Llama-3.1-8B. The x-axis shows the centers of 20 probability bins. The calibration curve (blue line with $\mu \pm \sigma$) displays actual accuracy per bin, while the gray shadow represents the probability proportion.

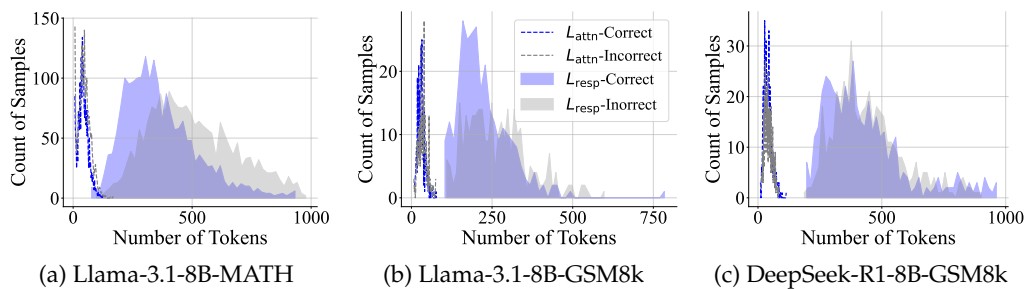

Figure 4: Histograms of response and attention chain lengths for correct & incorrect answers.

predicted probabilities concentrate around zero, leading to increased ECE. This highlights the need to control token count via similarity filtering to preserve a balanced confidence distribution. Conversely, $\widetilde{P}_{\mathcal{M}}$, developed from $\widetilde{P}_{\mathcal{M},\text{sim}}$, consistently outperforms $\widetilde{P}_{\mathcal{M},\text{sim}}$ in terms of AUROC, albeit at the expense of higher ECE. Thus, the optimal variant depends on the specific application requirements.

**Ablation Study** Since similarity filtering (8) effectively controls the size of the estimated attention space $\mathbb{S}$, and given that $L_{\text{attn}}$ inherently has higher similarity to $x_{\text{ans}}$ compared to $x_{\text{resp}}$ (Figure 7, appendix D.2), we investigate whether using only similarity scores without the attention backtracking described in § 4.1 is sufficient. Specifically, we set $x_{\text{attn}} = x_{\text{resp}}$ in § 4.2, ignoring § 4.1. Table 2 compares the generalizable results of this similarity-only variant against UQAC on smaller-scale models. The similarity-only variant performs notably worse than UQAC, exhibiting higher ECE and lower AUROC scores. This indicates that attention backtracking is crucial for effectively identifying semantically important tokens and enhancing calibration performance.

**Correlation Analysis** Previous studies such as Li et al. (2024a) report a negative correlation between a model's prediction accuracy and its UQ performance, suggesting that higher

|  |  | Llama-3.2-1B | | gemma-2-2b | | Qwen2.5-1.5B | | Average | |
|---|---|---|---|---|---|---|---|---|---|
|  |  | AUC ↑ | ECE ↓ | AUC ↑ | ECE ↓ | AUC ↑ | ECE ↓ | AUC ↑ | ECE ↓ |
| GSM8k | UQAC | 64.75 | 19.54 | 68.10 | 35.45 | 57.97 | 34.76 | 63.61 | 29.92 |
|  | w/o attns | 61.56 | 22.06 | 66.17 | 32.22 | 60.09 | 35.22 | 62.61 | 29.83 |
| MATH | UQAC | 67.92 | 20.65 | 74.09 | 15.23 | 71.71 | 31.45 | 71.24 | 22.45 |
|  | w/o attn | 65.25 | 24.55 | 70.30 | 18.94 | 72.65 | 31.57 | 69.40 | 25.02 |
| BBH | UQAC | 61.91 | 21.05 | 60.78 | 26.01 | 65.90 | 21.35 | 62.86 | 22.80 |
|  | w/o attn | 62.17 | 22.52 | 60.06 | 25.60 | 62.25 | 22.82 | 61.49 | 23.65 |

Table 2: Performance differences when applying similarity-based filtering (8) to all response tokens $x_{\text{resp}}$ instead of only the attention chain $L_{\text{attn}}$, *i.e.*, $x_{\text{attn}} = x_{\text{resp}}$.

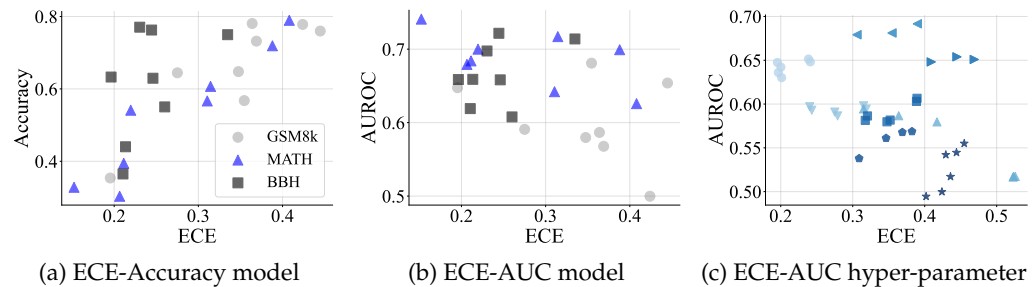

(a) ECE-Accuracy model      (b) ECE-AUC model      (c) ECE-AUC hyper-parameter

Figure 5: Correlation analysis of ECE with accuracy and AUROC for UQAC-$\widetilde{P}_{\mathcal{M}}$. In (a) and (b), different point types differentiate dataset; in (c) they distinguish models.

accuracy often coincides with poorer calibration. To verify whether this holds for autoregressive LLMs, Figure 5a plots each model's ECE against its accuracy across all datasets. The strong correlation shown in the figure, evidenced by an average Pearson's coefficient of 0.760, confirms that the conclusion still holds for LLMs. This is further discussed in appendix D.1 from the dataset perspective.

We further analyze the relationship between ECE and AUROC under two conditions. Across different models (presented by the same type of points), Figure 5b reveals a negligible correlation (average coefficient of $-0.147$), indicating that calibration and AUROC capture distinct performance aspects and should be evaluated jointly, echoing § 5. However, when examining $\widetilde{P}_{\mathcal{M}}$ with hyper-parameters, specifically, varying $L'_{\text{attn}}$ from 8 to 12 and the similarity threshold in (8) between 0 and 0.2, a more pronounced positive correlation (coefficient of 0.489) emerges. These findings suggest a trade-off between calibration and AUROC, warranting further exploration in future work.

**Hyper-Parameters** As introduced in § 5, we adopt a single, universal set of hyper-parameters for every model. This configuration yields stable, competitive results in most situations, yet task-specific tuning can sometimes improve calibration further.

Table 3 on the right presents a small hyper-parameter sweep for gemma-2-2b on BBH. Cells shaded in light blue correspond to our universal setting, while boldface highlights the best value achieved in each metric. Although our default choice falls slightly

| $K$ | $\theta$ | $L'_{\text{attn}}$ | AUROC (↑) | ECE (↓) |
|---|---|---|---|---|
| 8 | 0.5 | 10 | 58.11 ± 0.61 | 25.50 ± 0.52 |
| 8 | 0.7 | 10 | **59.32 ± 0.79** | 25.25 ± 0.38 |
| 16 | 0.5 | 10 | 56.59 ± 0.77 | 26.39 ± 0.96 |
| 16 | 0.5 | 16 | 59.25 ± 0.97 | **24.16 ± 0.79** |
| 16 | 0.7 | 10 | 58.10 ± 1.19 | 25.61 ± 0.82 |
| 32 | 0.5 | 10 | 57.18 ± 0.69 | 26.01 ± 0.65 |
| 32 | 0.7 | 10 | 58.83 ± 1.04 | 25.45 ± 0.68 |

Table 3: Hyper-parameter study with gemma-2-2b on BBH. $K$ is the number of attention heads (4), $\theta$ the attention weight threshold (6), and $L'_{\text{attn}}$ the number of tokens in the similarity-filtered token sequence $x'_{\text{attn}}$ (8).

short of the sweep's optimum, it remains a convenient, ready-to-use option that—as demonstrated in appendix D.4—delivers solid performance across a wide range of tasks.

# 7 Conclusion

This work addresses UQ for LLMs, particularly in tasks where final answers are derived from the intermediate reasoning steps. We propose UQAC, a method that leverages attention chains constructed according to the autoregressive attention weights to efficiently identify and track semantically critical tokens within the reasoning sequence, significantly reducing the space for uncertainty estimation. UQAC provides reliable uncertainty estimates with minimal computational overhead, requiring only a few parallel forward inferences without additional fine-tuning, multiple recurrent response sampling, or depending on external models. Empirical evaluations across diverse benchmarks and model architectures confirm that UQAC achieves superior calibration, especially when intermediate reasoning steps substantially influence final outcomes. Moreover, UQAC is applicable to any white-box autoregressive LLMs, preserves reasoning interpretability, and scales efficiently with increasing model size and complexity. Limitations are discussed in § 7.

## Limitations

Although UQAC is, in principle, compatible with any Transformer-based model that generates textual outputs, we do not explore tasks in the vision-only or multimodal domains. Due to resource constraints, our empirical analyses primarily focus on smaller LLMs. Theoretically, the conclusions in this paper should generalize to larger models, we have not empirically verified this assumption. UQAC may also underperform in certain cases, *e.g.*, gemma-2-9b on GSM8K, when run with its default hyper-parameters. Performance can be improved by tweaking the hyper-parameters, as shown in § 6, but such tuning deviates from our aim of providing a plug-and-play UQ method. Finally, although UQAC consistently achieves better calibration than baseline methods, its performance is still imperfect, leaving room for further improvement.

## Acknowledgments

This work was supported in part by NSF IIS-2008334, IIS-2106961, IIS-2403240, CAREER IIS-2144338, and ONR MURI N00014-17-1-2656. This research has benefitted from the Microsoft Accelerate Foundation Models Research (AFMR) grant program.

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

# A   Notations

Table 4 summarizes the notations and definitions included in this paper.

| Category | Notation | Definition |
|---|---|---|
| **Sets & Spaces** | $\mathbb{V}$ | Vocabulary set. |
| | $\mathbb{V}_{\text{stop}}$ | Set of stop words. |
| | $\mathbb{N}_{[1,N]}$ | Set of natural numbers from 1 to $N$. |
| | $\mathbb{S}$ | Reduced reasoning space. |
| | $\varnothing$ | Empty set. |
| **Tokens & Sequences** | $x$ | A sampled token. |
| | $\mathrm{x}$ | A random token variable. |
| | $\boldsymbol{x}$ | A sampled token sequence/vector. |
| | $\mathbf{x}$ | A random sequence variable. |
| | $\boldsymbol{x}_{\text{instr}}$ | Instruction sequence. |
| | $\boldsymbol{x}_{\text{resp}}$ | Response sequence. |
| | $\boldsymbol{x}_{\text{cot}}$ | Reasoning (CoT) sequence. |
| | $\boldsymbol{x}_{\text{ans}}$ | Answer token sequence. |
| | $\boldsymbol{x}_{\text{src}}^{(z)}$ | Source sequence/set at backtracking step $z$. |
| | $\boldsymbol{x}_{\text{tgt}}^{(z)}$ | Target sequence/set at backtracking step $z$. |
| | $\boldsymbol{x}_{\text{attn}}$ | Attention chain token sequence. |
| | $\boldsymbol{x}'_{\text{attn}}$ | Similarity-filtered attention chain sequence. |
| **Attention Bachtracking** | $\boldsymbol{\alpha}_T^{(l,h)}$ | Attention weights at step $T$ in layer $l$, head $h$. |
| | $\boldsymbol{\alpha}'$ | Re-weighted attention weights. |
| | $\boldsymbol{\alpha}_T^*$ | Aggregated attention weights at step $T$. |
| | $\gamma$ | Re-weighting factors. |
| | $\boldsymbol{\phi}$ | Cumulative attention weights from all source tokens. |
| | $\boldsymbol{w}$ | Similarity vector. |
| **Probabilistic Metrics** | $P$ | Exact probability. |
| | $\widetilde{P}$ | Approximated probability. |
| | $\overline{P}$ | Length-normalized approximated probability. |
| | $\mathcal{H}$ | Entropy measure. |
| **Numbers & Indices** | $L_{\text{instr}}$ | Length of the instruction sequence. |
| | $L_{\text{resp}}$ | Length of the response sequence. |
| | $L_{\text{cot}}$ | Length of the reasoning sequence. |
| | $L_{\text{ans}}$ | Length of the answer sequence. |
| | $L_{\text{attn}}$ | Length of the attention chain. |
| | $L'_{\text{attn}}$ | Length of the similarity-filtered attention chain. |
| | $L_{\text{tgt}}$ | Length of the target sequence/set. |
| | $t$ | Token index in a sequence. |
| | $T$ | Index of the latest predicted token during inference. |
| | $i, j, k$ | Generic indices (context-dependent). |
| | $l, h$ | Layer and head indices in model $\mathcal{M}$. |
| | $L, H$ | Total number of layers and heads in model $\mathcal{M}$. |
| | $z$ | Attention backtracking step index. |
| | $s, e$ | Position indices in source/target sequences or sets. |
| | $t_s, t_e$ | Specific token positions in source and target sequences. |
| | $C$ | Number of tokens selected for re-weighting. |
| | $\theta$ | Attention weight threshold. |
| | $\tau$ | Similarity threshold. |
| **Functions** | $f(\cdot)$ | Step-wise attention backtracking function. |
| | $\mathbb{I}(\cdot)$ | Indicator function. |
| | $\lvert \cdot \rvert$ | Cardinality of a set/sequence/vector. |
| | $\lVert \cdot \rVert$ | 2-norm of a vector. |
| | $\text{sim}(\cdot, \cdot)$ | Cosine similarity. |
| **Operators** | $\oplus$ | Concatenation of sequences. |
| | $\sqsubset$ | Proper subsequence relation (non-consecutive allowed). |
| | $\sqsubseteq$ | Subsequence relation. |
| | $\cdot^{\mathsf{T}}$ | Transpose of a vector or matrix. |
| | $\cdot^T$ | Exponentiation by $T$. |
| | $\cdot^{(T)}$ | Superscript $T$. |
| | $\arg\min_k \boldsymbol{a}$ | Index of the $k$-th smallest element in vector $\boldsymbol{a}$. |
| | $\arg\max_k \boldsymbol{a}$ | Index of the $k$-th largest element in vector $\boldsymbol{a}$. |
| | $\boldsymbol{a}_i$ | A vector from $\{\boldsymbol{a}_1, \boldsymbol{a}_2, \dots\}$ indexed by subscript $i$. |
| | $\boldsymbol{a}_{i,[j]}$ | $j$-th element in the vector $\boldsymbol{a}_i$. |
| | $\boldsymbol{a}_{i,[\backslash j]}$ | Subvector/subset of $\boldsymbol{a}_i$ excluding the $j$-th element. |
| | $\boldsymbol{a}_{i,[j,k]}$ | Subvector of $\boldsymbol{a}_i$ from the $j$-th to $k$-th elements (inclusive). |

Table 4: Summary of Notations and Definitions

## B UQAC Details

### B.1 Attention Re-Weighting

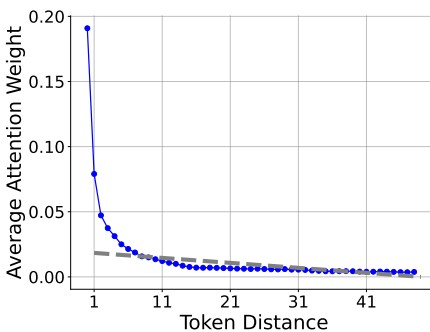

Figure 6: Averaged attention weights.

As discussed in § 4.1, to mitigate the inherent bias of LLMs toward emphasizing more recent context, we apply a sequence of decreasing re-weighting factors $\gamma \in [0, 1]^C$ to adjust the attention weights corresponding to the most recent $C$ tokens. To obtain these re-weighting factors, we first compute an average attention vector $\bar{\alpha}$. Specifically, we randomly select 1,000 samples from the GSM8k training partition and use Llama-3.2-1B to generate answers for each sample. We then average the attention weights from all heads and layers for the last 50 tokens in each generated response:

$$\bar{\alpha} \in [0,1]^{50} = \frac{1}{HLL_{\text{resp}}} \sum_{h=1}^{H} \sum_{l=1}^{L} \sum_{t=L_{\text{instr}}+1}^{L_{\text{instr}}+L_{\text{resp}}} \alpha_{t,[t-50:t]}^{(h,l)}. \tag{10}$$

To simplify subsequent analysis, we reverse the order of the computed average attention weights, placing the most recent token at the first position, i.e., $\bar{\alpha}_{[i]} \leftarrow \bar{\alpha}_{[50-i]}$. Next, we project $\bar{\alpha}$ into a two-dimensional space, with the $y$-axis representing attention weights and the $x$-axis representing token indices. We then fit a linear regression line that passes through the points $(C + 1, \bar{\alpha}_{[C+1]})$ and $(\lfloor (50 - C)/2 \rfloor, \bar{\alpha}_{[\lfloor (50-C)/2 \rfloor]})$. This regression line is defined as:

$$g(i) = \bar{\alpha}_{[C+1]} + \frac{\bar{\alpha}_{[\lfloor \frac{50-C}{2} \rfloor]} - \bar{\alpha}_{[C+1]}}{\lfloor \frac{50-C}{2} \rfloor - (C+1)} (i - (C+1)). \tag{11}$$

Subsequently, each re-weighting factor $\gamma_{[i]}$ is calculated as:

$$\gamma_{[C-i+1]} = \frac{g(i)}{\bar{\alpha}_{[i]}} \quad \forall i \in \mathbb{N}_{[1,C]}. \tag{12}$$

We emphasize that the resulting $\gamma$ values are approximations, as high precision is not critical for this adjustment. Consequently, we adopt these GSM8k and Llama-3.2-1B-derived factors universally across all models and datasets. In our experiments, we consistently set $C = 10$, yielding the following re-weighting factors:

$$\gamma = \begin{bmatrix} 0.93925344 \\ 0.87378443 \\ 0.81274293 \\ 0.73914525 \\ 0.67549127 \\ 0.59304059 \\ 0.46061748 \\ 0.32959151 \\ 0.20938152 \\ 0.16644488 \end{bmatrix}, \tag{13}$$

which is hard-coded in our implementation. The averaged attention weights and the fitted line are visualized in Figure 6.

### B.2 Attention Chain Construction

When transferring the target set $x_{\text{tgt}}$ to serve as the new source $x_{\text{src}}$ for the next iteration (i.e., updating $x_{\text{src}}^{(z)} = x_{\text{tgt}}^{(z)}$ for iteration $z + 1$), it is important to differentiate between *token*

*generation* and *token attention*. During token generation, a new token $x_{T+1}$ is produced based on the input token $x_T$, with the attention computation operating between $x_T$ and the preceding context $x_{\leqslant T}$. In contrast, during token attention, the projected key and value vectors are computed directly from the token $x_t$ as it is fed into the model.

Suppose that $x_{T+1}$ is identified as a semantically crucial token and we wish to trace back the tokens that contributed to its prediction via attention backtracking. In this case, we focus on the attention vectors generated during the token's *generation*. These vectors, computed with respect to the input token $x_T$, are defined by (2) (reproduced here for clarity):

$$\boldsymbol{\alpha}_T \in [0,1]^T = \text{softmax}\left(\frac{\boldsymbol{K}_T \boldsymbol{q}_T}{\sqrt{d_k}}\right). \tag{14}$$

Assume further that a token $x_t \in \boldsymbol{x}_{\text{tgt}}$ is the most attended token in the target set, *i.e.*,

$$\alpha_{T,[t]} = \max_i \alpha_{T,[i]} = \max_i \boldsymbol{k}_{T,i}^{\mathsf{T}} \boldsymbol{q}_T, \tag{15}$$

where $\boldsymbol{k}_{T,i} \triangleq \boldsymbol{K}_{T,[i]}$ denotes the $i$-th key vector in the key matrix $\boldsymbol{K}_T$ corresponding to token $x_i$. In this scenario, $x_t$ serves as a model input (generated from $x_{t-1}$) rather than an output.

Consequently, when propagating the target set to the next iteration, the indices are shifted back by one position $\boldsymbol{x}_{\text{src}}^{(z)} = \{x_{i-1} \mid x_i \in \boldsymbol{x}_{\text{tgt}}^{(z)}\}$. The same index-shifting principle applies to the initial source set $\boldsymbol{x}_{\text{src}}^{(0)} = \{x_{i-1} \mid x_i \in \boldsymbol{x}_{\text{ans}}\}$.

### B.3 Similarity-Based Filtering

Our similarity calculation (7) is practical and efficient as it leverages model $\mathcal{M}$'s hidden states $\boldsymbol{h}$ rather than external Sentence-BERT embeddings (Reimers & Gurevych, 2019; Kuhn et al., 2023; Lin et al., 2024b). Because these hidden states are produced "for free" during autoregressive decoding, no additional computational overhead is incurred. In contrast, extracting Sentence-BERT embeddings would require multiple extra forward passes through a separate model that employs a different vocabulary and tokenization scheme. Moreover, due to the autoregressive nature of $\mathcal{M}$, the hidden states corresponding to the same token at different positions are not necessarily identical:

$$\boldsymbol{h}_m \neq \boldsymbol{h}_n \quad \text{and} \quad 0 < \boldsymbol{h}_m^{\mathsf{T}} \boldsymbol{h}_n < 1, \quad \forall m, n \text{ s.t. } x_m = x_n, m \neq n, \tag{16}$$

This property helps reduce the risk of inadvertently filtering out short indicator tokens, such as "yes", "no", or multiple-choice options, when relying on external embeddings. Although several studies indicate that last-layer hidden states may be suboptimal for capturing semantics (Li & Li, 2024; Liu et al., 2024), our current configuration performs adequately in practice. We therefore leave a systematic search for improved semantic representations to future work.

## C Experiment Setup Details

### C.1 Datasets and Processing

We evaluate our approach using three widely recognized reasoning datasets that span various domains, including mathematical problem solving, logical reasoning, and commonsense reasoning. Only *test* partitions of the datasets are used for evaluation.

- **GSM8K** (Cobbe et al., 2021): A dataset comprising 8,792 (7,473 for training and 1,319 for test) high-quality grade school math word problems that require multi-step reasoning. Each problem includes a question and a detailed, step-by-step solution.
- **MATH** (Hendrycks et al., 2021): A collection of 12,500 (7,500 for training and 5,000 for test) challenging competition-level math problems covering subjects such as algebra, geometry, calculus, and more. Each problem is paired with a detailed solution.

- **BIG-Bench Hard (BBH)** (bench authors, 2023; Suzgun et al., 2023): A subset of BIG-Bench consisting of 23 tasks identified as particularly challenging for LLMs, summarizing to 6,511 test instances in total. These tasks span domains such as logical reasoning, mathematics, and commonsense reasoning. In addition to providing the correct answers, BBH includes detailed CoT reasoning annotations for each question, thereby enabling the evaluation of both final answers and intermediate reasoning processes.

For the mathematical reasoning tasks on GSM8K and MATH, we employ a *zero-shot* prompting strategy without additional formatting instructions, ensuring that models generate responses directly from the input questions. In most cases, the answers from all models are straightforwardly extractable. However, gemma-2 models require additional guidance; they are explicitly instructed to enclose their final answers within a \boxed{} wrapper to facilitate easier extraction. For BBH, we utilize 3 provided in-context examples, supplemented with explicit instructions and modified prompts. These modifications direct the models to encapsulate their final answers within the \boxed{} wrapper, ensuring consistent and reliable answer extraction across the dataset.

During our experiments, we set the maximum sequence length to 1,024 tokens for GSM8k and MATH, and to 1,536 tokens for BBH to accommodate the in-context examples in the latter dataset. For DeepSeek-R1, we further extend this limit by an additional 512 tokens to incorporate the "deep thinking" tokens.

Our implementation separates answer generation from the uncertainty quantification step. Once an answer is generated, we first extract and evaluate it to verify its correctness. If no answer is extracted, the instance is excluded from the UQ evaluation. Next, we subsample the instances to balance the number of correct and incorrect predictions, ensuring a reliable estimation of the calibration metrics. Specifically, if the number of correct predictions is lower than that of incorrect ones, we randomly sample a matching number of incorrect predictions. If both groups exceed 500 instances, we randomly select 500 instances from each group. Although this step does not affect the UQ scores, it does influence both the AUROC and ECE metrics. Therefore, we repeat the subsampling process five times and report the mean and standard deviation of the results.

## C.2 Baselines

In § 5, we compare the three variants of our proposed method, UQAC, namely $\widetilde{P}_{\mathcal{M},\text{attn}}$, $\widetilde{P}_{\mathcal{M},\text{sim}}$, and $\widetilde{P}_{\mathcal{M}}$, against a diverse set of baselines. These baselines are selected to cover a broad spectrum of uncertainty quantification techniques, incorporating both token-level and response-level approaches. In particular, we consider the following methods:

1. $P_{\mathcal{M}}(x_{\text{ans}}|x_{\text{cot}}, x_{\text{instr}})$: The joint conditional probability of the answer, conditioned on both the CoT reasoning and sequence and the instruction sequence.
2. $P_{\mathcal{M}}(x_{\text{resp}}|x_{\text{instr}})$: The joint conditional probability of the entire response given the instruction.
3. $\overline{P}_{\mathcal{M}}(x_{\text{ans}})$: The mean conditional probability computed over the answer tokens.
4. $\overline{P}_{\mathcal{M}}(x_{\text{resp}})$: The mean conditional probability computed over all tokens in the response.
5. **Predictive Entropy** $\mathcal{H}$ (Kuhn et al., 2023): A token-level uncertainty measure that aggregates entropy over the response, where the value do not have a fixed range:

$$\mathcal{H} \in [0, L_{\text{resp}}] = -\sum_{t=L_{\text{instr}}+1}^{L_{\text{instr}}+L_{\text{resp}}} \sum_{x_t \sim \mathbb{V}} P_{\mathcal{M}}(x_t|x_{<t}) \log P_{\mathcal{M}}(x_t|x_{<t}). \tag{17}$$

6. **Length-Normalized Predictive Entropy** $\overline{\mathcal{H}}$ (Malinin & Gales, 2021): It normalizes the predictive entropy by the response length to account for variations in output size:

$$\overline{\mathcal{H}} \in [0, 1] = \frac{\mathcal{H}}{L_{\text{resp}}}. \tag{18}$$

7. **Self-Consistency** (Wang et al., 2023): The probability averaged over 5 independently sampled answers, thereby reflecting the consistency of the model's outputs. We use a temperature of 0.5 to balance the samples' accuracy and diversity.

8. **Verbalized Uncertainty** (Xiong et al., 2024): A model-generated confidence score obtained via additional prompting.

The first six baselines derive their uncertainty estimates directly from token probabilities, while the latter two rely on aggregating information from multiple or additional model outputs to capture uncertainty at a higher level. It is important to note that earlier approaches, such as the semantic entropy methods proposed in (Kuhn et al., 2023; Lin et al., 2024b), provide only a weaker approximation of Self-Consistency. These methods focus on capturing lexical diversity but do so at the expense of substantially higher computational overhead. In addition, the semantic ambiguity are minimal in our reasoning tasks, making these methods less relevant. Consequently, we neglect such methods.

In appendix E, we report additional experimental results for several UQAC variants and baselines that were not included in the main discussion. These additional methods, although derived from our primary baselines, possess less rigorous theoretical foundations and exhibit inferior performance in our experiments. They are detailed below:

9. **Averaged Attention Approximation** UQAC-$\overline{P}_{\mathcal{M},\text{attn}}$: This variant approximates the predictive distribution using attention scores, which are averaged over all tokens in the attention chain $x_{\text{attn}}$ (see § 4.3).
10. **Averaged Similarity Approximation** UQAC-$\overline{P}_{\mathcal{M},\text{sim}}$: Similar to the previous variant, this method employs similarity-based approximations of the predictive distribution, averaging over tokens in the similarity-filtered attention chain $x'_{\text{attn}}$ (refer to § 4.3).
11. **Predictive Answer Entropy** $\mathcal{H}(x_{\text{ans}})$: This metric is analogous to the predictive entropy but is calculated exclusively over answer tokens. It is defined as

$$\mathcal{H}(x_{\text{ans}}) = -\sum_{t=L_{\text{instr}}+L_{\text{cot}}+1}^{L_{\text{instr}}+L_{\text{resp}}} \sum_{x_t \sim \mathbb{V}} P_{\mathcal{M}}(x_t|x_{<t}) \log P_{\mathcal{M}}(x_t|x_{<t}). \tag{19}$$

12. **Predictive Answer Length-Normalized Entropy** $\overline{\mathcal{H}}(x_{\text{ans}})$: This is the length-normalized version of the predictive answer entropy, computed as

$$\overline{\mathcal{H}}(x_{\text{ans}}) = \frac{\mathcal{H}(x_{\text{ans}})}{L_{\text{ans}}}, \tag{20}$$

where $L_{\text{ans}}$ denotes the length of the answer.

In our results tables (*e.g.*, Table 1), we present the full mathematical expressions for baselines that have a robust theoretical basis and are computed exactly as defined. For methods lacking such foundations, we adopt a shorthand notation (*e.g.*, $\overline{P}_{\mathcal{M}}(x_{\text{ans}})$ or $\overline{P}_{\mathcal{M},\text{attn}}$) for clarity. All reported entropy metrics are computed using $1 - \mathcal{H}$, as shown in Table 1.

### C.3 Evaluation Metrics

**AUROC** The receiver operating characteristic area under the curve (AUROC) is a widely used metric in binary classification tasks. The ROC curve is constructed by plotting the true positive rate (TPR, also known as *recall*) against the false positive rate (FPR) for varying decision thresholds $t \in (0,1)$. Given the counts of true positives (TP), true negatives (TN), false positives (FP), and false negatives (FN), the TPR and FPR are computed as

$$\text{TPR} = \frac{\text{TP}}{\text{TP} + \text{FN}}, \qquad \text{FPR} = \frac{\text{FP}}{\text{FP} + \text{TN}}. \tag{21}$$

The AUROC represents the probability that a randomly selected positive instance is ranked higher than a randomly selected negative instance. Formally, it is defined as

$$\text{ROC-AUC} = \int_0^1 \text{TPR}(t) \frac{d}{dt} \text{FPR}(t) \, dt, \tag{22}$$

and in practice, it is often approximated using numerical integration methods.[1]

---

[1]We use scikit-learn's `roc_auc_score` function in our implementation.

While AUROC is effective in evaluating the discriminative ability of a classifier, *i.e.* its capacity to separate correct from incorrect predictions, it does not assess the calibration of the predicted confidence scores. Calibration refers to the degree of agreement between the predicted probabilities and the actual likelihoods of the outcomes. For instance, consider a scenario with three instances having predicted confidence scores of 0.9, 0.5, and 0.1, where the first two predictions are correct and the third is incorrect. In this case, AUROC would yield a score of 1, indicating perfect separation. However, the same AUROC score of 1 would result even if the predictions were assigned extremely low and uninformative confidence values (*e.g.*, $9 \times 10^{-3}$, $8 \times 10^{-10}$, and $7.99 \times 10^{-10}$), despite the fact that such scores lack meaningful interpretation. Consequently, a high AUROC does not guarantee that the confidence estimates reflect the true probability of correctness. If a model outputs a maximum estimated confidence of only $1 \times 10^{-3}$ across 10,000 instances, users might be misled about the model's overall certainty despite an ostensibly perfect AUROC.

**ECE**   To address this limitation, we also incorporate calibration metrics into our evaluation, with a particular focus on the Expected Calibration Error (ECE; Naeini et al., 2015). ECE quantifies the average discrepancy between predicted probabilities and the corresponding empirical frequencies. In the binary classification setting, the set of predicted probabilities $\{\hat{p}_n\}_{n=1}^{N}$ is partitioned into $S$ equal-width bins (with $S = 20$ in our experiments). Formally, for each bin $s \in \{1, \ldots, S\}$, we define

$$\mathbb{B}_s = \{n \in \{1, \ldots, N\} \mid \hat{p}_n \in (\rho_s, \rho_{s+1}]\}. \tag{23}$$

The ECE is then calculated as

$$\text{ECE} = \sum_{s=1}^{S} \frac{|\mathbb{B}_s|}{N} \left| \text{acc}(\mathbb{B}_s) - \text{conf}(\mathbb{B}_s) \right|,$$

$$\text{acc}(\mathbb{B}_s) = \frac{1}{|\mathbb{B}_s|} \sum_{n \in \mathbb{B}_s} y_n, \qquad \text{conf}(\mathbb{B}_s) = \frac{1}{|\mathbb{B}_s|} \sum_{n \in \mathbb{B}_s} \hat{p}_n. \tag{24}$$

In this formulation, $\text{acc}(\mathbb{B}_s)$ represents the empirical accuracy within bin $s$ and $\text{conf}(\mathbb{B}_s)$ denotes the average predicted confidence. Here, $|\mathbb{B}_s|$ is the number of instances in bin $s$ (while the notation $|\cdot|$ denotes the absolute value when applied to real numbers). By quantifying the calibration error, ECE provides critical insights into how well the predicted confidence scores align with actual prediction accuracy, thereby complementing the discriminative evaluation offered by AUROC.

## D   Result Discussion

### D.1   Potential Overfitting

On GSM8k, nearly all uncertainty quantification methods show a significant performance gap compared to those on more challenging datasets. As illustrated in Figure 3f versus Figures 3d and 3e, GSM8k exhibits a marked overconfidence: the probability distribution is highly centralized around 1 and the calibration curve deviates substantially from the ideal diagonal. This behavior suggests that state-of-the-art LLMs may be overfitting on GSM8k, thereby questioning its reliability as a benchmark for evaluating LLM capabilities.

### D.2   Response Lengths

Figure 4 illustrates the lengths of the entire response sequences $L_{\text{resp}}$ and the attention chain $L_{\text{attn}}$, categorized by answer correctness. The lengths of the similarity-filtered chain ($L'_{\text{attn}}$) are omitted as they are controlled. Across all models, $L_{\text{attn}}$ is less than 10% of $L_{\text{resp}}$, demonstrating that the attention backtracking function $f$ effectively reduces computational overhead. Additionally, while $L_{\text{resp}}$ is noticeably longer for incorrect predictions, this difference is less pronounced for $L_{\text{attn}}$. Given that tokens in the attention chain have a higher average similarity to the answer tokens (shown in Figure 7 with $\text{sim}(x_{\text{ans}}, x_{\text{attn}}) = 0.257$ versus $\text{sim}(x_{\text{ans}}, x_{\text{resp}}) = 0.206$), these results confirm that $f$ successfully captures the reasoning process and identifies critical tokens.

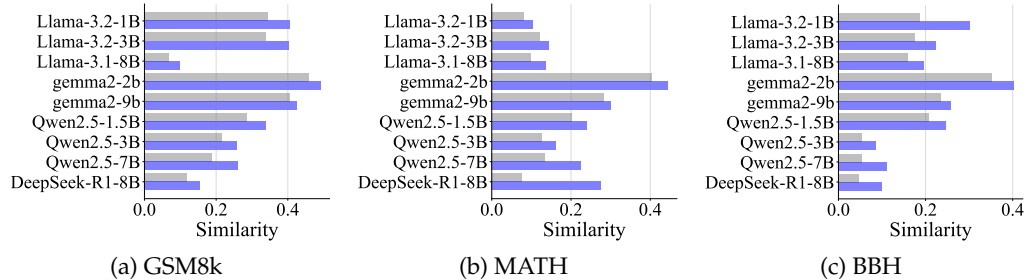

|                | (a) GSM8k | (b) MATH | (c) BBH |
|----------------|-----------|----------|---------|

Figure 7: Average token similarity between the answer tokens $x_{ans}$ and 1) the attention chain tokens $x_{attn}$ (blue bars) and 2) the response tokens $x_{resp}$ (gray bars). The similarity scores are computed by averaging the cosine similarities of token embeddings over all examples. It shows that $x_{attn}$ tokens are inherently more similar to $x_{ans}$ tokens than average $x_{resp}$ tokens, indicating that the attention chain is more likely to capture the answer-relevant context.

### D.3 Computational Overhead

Figure 2 illustrates the average computational overhead (excluding I/O) measured on the Llama-3.1-8B model when generating outputs of approximately 500 tokens. Here, "Infer" refers to the baseline inference time needed to generate responses; "VU" denotes the Verbalized Uncertainty approach; and "SC" indicates Self-Consistency with 5 sampled responses. Compared to standard inference, UQAC introduces only a slight computational overhead, demonstrating higher efficiency compared to the Verbalized Uncertainty method. In contrast, although Self-Consistency achieves superior calibration performance, it substantially increases computational requirements. This overhead makes Self-Consistency less feasible for real-time applications, particularly in scenarios demanding extensive output generation or strict latency constraints.

### D.4 Hyper-Parameters

Tables 5–9 present additional experiments that analyze how various hyper-parameters affect UQAC's performance. Specifically, $K$ denotes the number of attention heads in (4); $\theta$ is the attention-weight threshold in (6); $L'_{attn}$ is the length of the similarity-filtered token sequence $x'_{attn}$, as defined in (8); and $\tau$ represents the similarity threshold. The configuration highlighted in light blue corresponds to the setup reported in the main paper.

| $K$ | $\theta$ | $L'_{attn}$ | $\tau$ | AUROC | ECE |
|-----|----------|-------------|--------|-------|-----|
| 8   | 0.3      | 10          | 0.0    | $65.38 \pm 1.05$ | $18.08 \pm 0.75$ |
| 8   | 0.5      | 10          | 0.0    | $65.09 \pm 1.12$ | $18.62 \pm 0.98$ |
| 8   | 0.7      | 10          | 0.0    | $65.61 \pm 1.08$ | $17.64 \pm 0.82$ |
| 16  | 0.3      | 10          | 0.0    | $64.42 \pm 1.21$ | $18.71 \pm 1.07$ |
| 16  | 0.5      | 10          | 0.0    | $64.15 \pm 1.12$ | $18.88 \pm 1.02$ |
| 16  | 0.7      | 10          | 0.0    | $64.62 \pm 1.38$ | $19.10 \pm 1.11$ |
| 32  | 0.3      | 10          | 0.0    | $63.88 \pm 1.45$ | $19.52 \pm 1.37$ |
| 32  | 0.5      | 10          | 0.0    | $63.76 \pm 1.33$ | $19.24 \pm 1.24$ |
| 32  | 0.7      | 10          | 0.0    | $63.30 \pm 1.45$ | $19.32 \pm 1.38$ |
| 16  | 0.5      | 8           | 0.0    | $65.14 \pm 1.04$ | $23.85 \pm 0.97$ |
| 16  | 0.5      | 8           | 0.2    | $64.81 \pm 1.00$ | $24.10 \pm 0.96$ |
| 16  | 0.5      | 8           | 0.5    | $61.90 \pm 1.01$ | $31.33 \pm 0.91$ |
| 16  | 0.5      | 10          | 0.2    | $64.21 \pm 1.22$ | $20.02 \pm 1.30$ |
| 16  | 0.5      | 10          | 0.5    | $61.68 \pm 1.13$ | $29.47 \pm 1.16$ |
| 16  | 0.5      | 12          | 0.0    | $63.60 \pm 1.12$ | $19.67 \pm 0.95$ |
| 16  | 0.5      | 12          | 0.2    | $63.01 \pm 1.14$ | $20.12 \pm 0.91$ |
| 16  | 0.5      | 12          | 0.5    | $61.55 \pm 1.08$ | $29.32 \pm 1.01$ |

Table 5: Hyper-parameter study of Llama-3.2-1B on GSM8K.

| $K$ | $\theta$ | $L'_{\text{attn}}$ | $\tau$ | AUROC | ECE |
|---|---|---|---|---|---|
| 8 | 0.3 | 10 | 0.0 | $58.30 \pm 0.70$ | $34.90 \pm 0.34$ |
| 8 | 0.5 | 10 | 0.0 | $58.68 \pm 0.87$ | $34.85 \pm 0.22$ |
| 8 | 0.7 | 10 | 0.0 | $59.28 \pm 0.51$ | $34.62 \pm 0.23$ |
| 16 | 0.3 | 10 | 0.0 | $57.73 \pm 0.55$ | $34.98 \pm 0.30$ |
| 16 | 0.5 | 10 | 0.0 | $57.53 \pm 0.60$ | $35.11 \pm 0.28$ |
| 16 | 0.7 | 10 | 0.0 | $58.27 \pm 0.41$ | $34.94 \pm 0.20$ |
| 32 | 0.3 | 10 | 0.0 | $58.16 \pm 1.01$ | $35.38 \pm 0.05$ |
| 32 | 0.5 | 10 | 0.0 | $58.62 \pm 0.98$ | $35.69 \pm 0.25$ |
| 32 | 0.7 | 10 | 0.0 | $58.57 \pm 0.82$ | $34.99 \pm 0.43$ |
| 16 | 0.5 | 8 | 0.0 | $60.30 \pm 0.40$ | $38.86 \pm 0.37$ |
| 16 | 0.5 | 8 | 0.2 | $60.66 \pm 0.49$ | $38.93 \pm 0.34$ |
| 16 | 0.5 | 8 | 0.5 | $48.07 \pm 0.64$ | $49.87 \pm 0.85$ |
| 16 | 0.5 | 10 | 0.2 | $58.17 \pm 0.73$ | $35.20 \pm 0.32$ |
| 16 | 0.5 | 10 | 0.5 | $47.75 \pm 0.55$ | $47.87 \pm 0.66$ |
| 16 | 0.5 | 12 | 0.0 | $58.13 \pm 0.71$ | $31.74 \pm 0.56$ |
| 16 | 0.5 | 12 | 0.2 | $58.64 \pm 0.79$ | $32.05 \pm 0.64$ |
| 16 | 0.5 | 12 | 0.5 | $47.65 \pm 0.57$ | $46.08 \pm 0.64$ |

Table 6: Hyper-parameter study for Qwen2.5-1.5B on GSM8K.

| $K$ | $\theta$ | $L'_{\text{attn}}$ | $\tau$ | AUROC | ECE |
|---|---|---|---|---|---|
| 8 | 0.3 | 10 | 0.0 | $68.36 \pm 0.68$ | $35.77 \pm 0.43$ |
| 8 | 0.5 | 10 | 0.0 | $68.43 \pm 0.61$ | $35.72 \pm 0.40$ |
| 8 | 0.7 | 10 | 0.0 | $68.80 \pm 0.52$ | $35.94 \pm 0.33$ |
| 16 | 0.3 | 10 | 0.0 | $68.30 \pm 1.06$ | $35.28 \pm 0.39$ |
| 16 | 0.5 | 10 | 0.0 | $68.10 \pm 0.99$ | $35.45 \pm 0.47$ |
| 16 | 0.7 | 10 | 0.0 | $67.99 \pm 0.93$ | $35.68 \pm 0.34$ |
| 32 | 0.3 | 10 | 0.0 | $69.24 \pm 0.99$ | $34.70 \pm 0.26$ |
| 32 | 0.5 | 10 | 0.0 | $68.99 \pm 0.93$ | $34.92 \pm 0.21$ |
| 32 | 0.7 | 10 | 0.0 | $69.19 \pm 0.75$ | $35.05 \pm 0.11$ |
| 16 | 0.5 | 8 | 0.0 | $69.14 \pm 0.77$ | $38.97 \pm 0.35$ |
| 16 | 0.5 | 8 | 0.2 | $69.18 \pm 0.76$ | $38.99 \pm 0.36$ |
| 16 | 0.5 | 8 | 0.5 | $67.92 \pm 0.68$ | $40.32 \pm 0.36$ |
| 16 | 0.5 | 10 | 0.2 | $68.14 \pm 0.97$ | $35.47 \pm 0.49$ |
| 16 | 0.5 | 10 | 0.5 | $66.70 \pm 0.72$ | $37.78 \pm 0.32$ |
| 16 | 0.5 | 12 | 0.0 | $67.89 \pm 0.93$ | $30.59 \pm 0.44$ |
| 16 | 0.5 | 12 | 0.2 | $67.94 \pm 0.91$ | $30.60 \pm 0.47$ |
| 16 | 0.5 | 12 | 0.5 | $66.43 \pm 0.63$ | $33.98 \pm 0.26$ |

Table 7: Hyper-parameter study for gemma-2-2b on GSM8K.

| $K$ | $\theta$ | $L'_{\text{attn}}$ | $\tau$ | AUROC | ECE |
|---|---|---|---|---|---|
| 8 | 0.3 | 10 | 0.0 | $73.31 \pm 0.64$ | $15.77 \pm 0.71$ |
| 8 | 0.5 | 10 | 0.0 | $73.56 \pm 0.67$ | $15.01 \pm 0.83$ |
| 8 | 0.7 | 10 | 0.0 | $73.87 \pm 0.73$ | $14.19 \pm 0.82$ |
| 16 | 0.3 | 10 | 0.0 | $73.92 \pm 1.07$ | $15.68 \pm 0.56$ |
| 16 | 0.5 | 10 | 0.0 | $74.08 \pm 1.19$ | $15.23 \pm 0.73$ |
| 16 | 0.7 | 10 | 0.0 | $73.97 \pm 1.16$ | $14.76 \pm 0.95$ |
| 32 | 0.3 | 10 | 0.0 | $74.22 \pm 0.90$ | $14.62 \pm 0.64$ |
| 32 | 0.5 | 10 | 0.0 | $74.41 \pm 0.84$ | $14.59 \pm 0.45$ |
| 32 | 0.7 | 10 | 0.0 | $74.02 \pm 0.85$ | $13.86 \pm 0.44$ |
| 16 | 0.5 | 8 | 0.0 | $74.86 \pm 0.96$ | $17.25 \pm 0.41$ |
| 16 | 0.5 | 12 | 0.0 | $74.18 \pm 1.16$ | $14.11 \pm 0.93$ |

Table 8: Hyper-parameter study for gemma-2-2b on MATH.

| $K$ | $\theta$ | $L'_{\text{attn}}$ | $\tau$ | AUROC | ECE |
|---|---|---|---|---|---|
| 8 | 0.3 | 10 | 0.0 | $57.60 \pm 1.01$ | $25.76 \pm 0.50$ |
| 8 | 0.5 | 10 | 0.0 | $58.11 \pm 0.61$ | $25.50 \pm 0.52$ |
| 8 | 0.7 | 10 | 0.0 | $59.32 \pm 0.79$ | $25.25 \pm 0.38$ |
| 16 | 0.3 | 10 | 0.0 | $56.43 \pm 0.81$ | $26.76 \pm 0.84$ |
| 16 | 0.5 | 10 | 0.0 | $56.59 \pm 0.77$ | $26.39 \pm 0.96$ |
| 16 | 0.7 | 10 | 0.0 | $58.10 \pm 1.19$ | $25.61 \pm 0.82$ |
| 32 | 0.3 | 10 | 0.0 | $56.73 \pm 0.65$ | $26.64 \pm 0.65$ |
| 32 | 0.5 | 10 | 0.0 | $57.18 \pm 0.69$ | $26.01 \pm 0.65$ |
| 32 | 0.7 | 10 | 0.0 | $58.83 \pm 1.04$ | $25.45 \pm 0.68$ |
| 16 | 0.5 | 12 | 0.0 | $56.63 \pm 1.05$ | $25.87 \pm 0.97$ |
| 16 | 0.5 | 16 | 0.0 | $59.25 \pm 0.97$ | $24.16 \pm 0.79$ |

Table 9: Hyper-parameter study for Gemma-2-2b on BBH.

# E   Complete Results

Tables 10–36 present the complete results for all experiments in the main paper.

| | | AUROC | ECE |
|---|---|---|---|
| Token Probability | $\bar{P}_{\mathcal{M}}(x_{\text{ans}})$ | $62.85 \pm 0.31$ | $48.93 \pm 0.14$ |
| | $\bar{P}_{\mathcal{M}}(x_{\text{resp}})$ | $65.47 \pm 0.80$ | $36.81 \pm 0.05$ |
| | $P_{\mathcal{M}}(x_{\text{ans}}|x_{\text{cot}}, x_{\text{instr}})$ | $63.62 \pm 0.32$ | $48.47 \pm 0.15$ |
| | $P_{\mathcal{M}}(x_{\text{resp}}|x_{\text{instr}})$ | $68.71 \pm 0.41$ | $50.00 \pm 0.00$ |
| Token Entropy | $\overline{\mathcal{H}}$ | $65.27 \pm 0.95$ | $20.22 \pm 0.13$ |
| | $\overline{\mathcal{H}}(x_{\text{ans}})$ | $62.65 \pm 0.27$ | $47.56 \pm 0.28$ |
| | $\mathcal{H}$ | $67.54 \pm 0.74$ | - |
| | $\mathcal{H}(x_{\text{ans}})$ | $63.46 \pm 0.29$ | - |
| Multi-Round Prompting | Verbalized Uncertainty | $53.75 \pm 0.57$ | $30.85 \pm 0.43$ |
| | Self-Consistency | $72.77 \pm 0.95$ | $14.21 \pm 0.77$ |
| UQAC | $\bar{P}_{\mathcal{M},\text{attn}}$ | $62.40 \pm 1.23$ | $38.49 \pm 0.11$ |
| | $\bar{P}_{\mathcal{M},\text{sim}}$ | $63.04 \pm 1.20$ | $42.35 \pm 0.14$ |
| | $\widetilde{P}_{\mathcal{M},\text{attn}}$ | $61.39 \pm 0.88$ | $44.01 \pm 0.21$ |
| | $\widetilde{P}_{\mathcal{M},\text{sim}}$ | $63.33 \pm 1.17$ | $20.95 \pm 0.62$ |
| | $\widetilde{P}_{\mathcal{M}}$ | $64.75 \pm 1.19$ | $19.54 \pm 1.27$ |

Table 10: Llama-3.2-1B-Instruct results on the GSM8k dataset.

| | | AUROC | ECE |
|---|---|---|---|
| Token Probability | $\overline{P}_{\mathcal{M}}(x_{\mathrm{ans}})$ | $59.19 \pm 0.74$ | $49.23 \pm 0.05$ |
| | $\overline{P}_{\mathcal{M}}(x_{\mathrm{resp}})$ | $64.03 \pm 0.62$ | $38.79 \pm 0.03$ |
| | $P_{\mathcal{M}}(x_{\mathrm{ans}}|x_{\mathrm{cot}}, x_{\mathrm{instr}})$ | $59.76 \pm 0.74$ | $49.06 \pm 0.05$ |
| | $P_{\mathcal{M}}(x_{\mathrm{resp}}|x_{\mathrm{instr}})$ | $68.75 \pm 0.51$ | $50.00 \pm 0.00$ |
| Token Entropy | $\overline{\mathcal{H}}$ | $64.72 \pm 0.54$ | $24.18 \pm 0.07$ |
| | $\overline{\mathcal{H}}(x_{\mathrm{ans}})$ | $60.00 \pm 0.78$ | $48.25 \pm 0.05$ |
| | $\mathcal{H}$ | $68.22 \pm 0.67$ | - |
| | $\mathcal{H}(x_{\mathrm{ans}})$ | $60.55 \pm 0.77$ | - |
| Multi-Round Prompting | Verbalized Uncertainty | $63.31 \pm 0.40$ | $38.60 \pm 0.22$ |
| | Self-Consistency | $70.35 \pm 0.76$ | $28.80 \pm 0.35$ |
| UQAC | $\overline{P}_{\mathcal{M},\mathrm{attn}}$ | $61.99 \pm 0.62$ | $41.08 \pm 0.04$ |
| | $\overline{P}_{\mathcal{M},\mathrm{sim}}$ | $59.06 \pm 0.52$ | $44.50 \pm 0.05$ |
| | $\widetilde{P}_{\mathcal{M},\mathrm{attn}}$ | $60.94 \pm 0.74$ | $38.55 \pm 0.58$ |
| | $\widetilde{P}_{\mathcal{M},\mathrm{sim}}$ | $58.85 \pm 0.50$ | $22.80 \pm 0.47$ |
| | $\widetilde{P}_{\mathcal{M}}$ | $59.08 \pm 0.55$ | $27.51 \pm 0.20$ |

Table 11: Llama-3.2-3B-Instruct results on the GSM8k dataset.

| | | AUROC | ECE |
|---|---|---|---|
| Token Probability | $\overline{P}_{\mathcal{M}}(x_{\mathrm{ans}})$ | $68.71 \pm 1.12$ | $49.64 \pm 0.02$ |
| | $\overline{P}_{\mathcal{M}}(x_{\mathrm{resp}})$ | $61.14 \pm 1.10$ | $41.65 \pm 0.05$ |
| | $P_{\mathcal{M}}(x_{\mathrm{ans}}|x_{\mathrm{cot}}, x_{\mathrm{instr}})$ | $68.90 \pm 1.11$ | $49.47 \pm 0.03$ |
| | $P_{\mathcal{M}}(x_{\mathrm{resp}}|x_{\mathrm{instr}})$ | $68.17 \pm 0.98$ | $49.99 \pm 0.00$ |
| Token Entropy | $\overline{\mathcal{H}}$ | $61.07 \pm 1.09$ | $30.12 \pm 0.11$ |
| | $\overline{\mathcal{H}}(x_{\mathrm{ans}})$ | $72.10 \pm 1.24$ | $49.32 \pm 0.08$ |
| | $\mathcal{H}$ | $67.59 \pm 1.05$ | - |
| | $\mathcal{H}(x_{\mathrm{ans}})$ | $72.30 \pm 1.23$ | - |
| Multi-Round Prompting | Verbalized Uncertainty | $62.27 \pm 0.48$ | $45.24 \pm 0.10$ |
| | Self-Consistency | $67.53 \pm 1.04$ | $37.01 \pm 0.44$ |
| UQAC | $\overline{P}_{\mathcal{M},\mathrm{attn}}$ | $53.35 \pm 1.11$ | $42.08 \pm 0.09$ |
| | $\overline{P}_{\mathcal{M},\mathrm{sim}}$ | $57.72 \pm 1.54$ | $47.45 \pm 0.06$ |
| | $\widetilde{P}_{\mathcal{M},\mathrm{attn}}$ | $52.41 \pm 1.42$ | $41.25 \pm 1.19$ |
| | $\widetilde{P}_{\mathcal{M},\mathrm{sim}}$ | $57.64 \pm 1.58$ | $31.16 \pm 0.56$ |
| | $\widetilde{P}_{\mathcal{M}}$ | $58.66 \pm 1.03$ | $36.39 \pm 0.22$ |

Table 12: Meta-Llama-3.1-8B-Instruct results on the GSM8k dataset.

|  |  | AUROC | ECE |
|---|---|---|---|
| Token Probability | $\bar{P}_{\mathcal{M}}(x_{\text{ans}})$ | $61.54 \pm 0.44$ | $49.60 \pm 0.03$ |
| | $\bar{P}_{\mathcal{M}}(x_{\text{resp}})$ | $70.09 \pm 0.29$ | $42.23 \pm 0.04$ |
| | $P_{\mathcal{M}}(x_{\text{ans}}|x_{\text{cot}}, x_{\text{instr}})$ | $60.20 \pm 0.43$ | $49.14 \pm 0.04$ |
| | $P_{\mathcal{M}}(x_{\text{resp}}|x_{\text{instr}})$ | $73.24 \pm 0.78$ | $49.99 \pm 0.00$ |
| Token Entropy | $\bar{\mathcal{H}}$ | $71.47 \pm 0.42$ | $33.54 \pm 0.11$ |
| | $\bar{\mathcal{H}}(x_{\text{ans}})$ | $61.54 \pm 0.44$ | $48.84 \pm 0.07$ |
| | $\mathcal{H}$ | $73.22 \pm 0.77$ | - |
| | $\mathcal{H}(x_{\text{ans}})$ | $60.05 \pm 0.42$ | - |
| Multi-Round Prompting | Verbalized Uncertainty | $58.14 \pm 0.69$ | $41.91 \pm 0.09$ |
| | Self-Consistency | $38.75 \pm 10.00$ | $34.13 \pm 3.36$ |
| UQAC | $\bar{P}_{\mathcal{M},\text{attn}}$ | $65.71 \pm 1.10$ | $44.52 \pm 0.04$ |
| | $\bar{P}_{\mathcal{M},\text{sim}}$ | $66.31 \pm 1.02$ | $46.94 \pm 0.04$ |
| | $\widetilde{P}_{\mathcal{M},\text{attn}}$ | $65.76 \pm 0.79$ | $32.48 \pm 0.26$ |
| | $\widetilde{P}_{\mathcal{M},\text{sim}}$ | $65.83 \pm 1.02$ | $27.70 \pm 0.84$ |
| | $\widetilde{P}_{\mathcal{M}}$ | $68.10 \pm 0.99$ | $35.45 \pm 0.47$ |

Table 13: gemma-2-2b-it results on the GSM8k dataset.

|  |  | AUROC | ECE |
|---|---|---|---|
| Token Probability | $\bar{P}_{\mathcal{M}}(x_{\text{ans}})$ | $50.35 \pm 0.61$ | $49.90 \pm 0.01$ |
| | $\bar{P}_{\mathcal{M}}(x_{\text{resp}})$ | $55.62 \pm 1.55$ | $45.51 \pm 0.05$ |
| | $P_{\mathcal{M}}(x_{\text{ans}}|x_{\text{cot}}, x_{\text{instr}})$ | $49.45 \pm 0.71$ | $49.92 \pm 0.03$ |
| | $P_{\mathcal{M}}(x_{\text{resp}}|x_{\text{instr}})$ | $60.31 \pm 0.59$ | $50.00 \pm 0.00$ |
| Token Entropy | $\bar{\mathcal{H}}$ | $59.60 \pm 1.43$ | $41.62 \pm 0.08$ |
| | $\bar{\mathcal{H}}(x_{\text{ans}})$ | $43.75 \pm 0.52$ | $49.85 \pm 0.02$ |
| | $\mathcal{H}$ | $64.90 \pm 1.25$ | - |
| | $\mathcal{H}(x_{\text{ans}})$ | $43.01 \pm 0.43$ | - |
| Multi-Round Prompting | Verbalized Uncertainty | $51.38 \pm 0.08$ | $49.75 \pm 0.01$ |
| | Self-Consistency | $53.45 \pm 1.82$ | $31.82 \pm 1.35$ |
| UQAC | $\bar{P}_{\mathcal{M},\text{attn}}$ | $57.16 \pm 0.94$ | $46.81 \pm 0.02$ |
| | $\bar{P}_{\mathcal{M},\text{sim}}$ | $66.76 \pm 1.12$ | $48.59 \pm 0.02$ |
| | $\widetilde{P}_{\mathcal{M},\text{attn}}$ | $58.64 \pm 0.77$ | $26.15 \pm 0.56$ |
| | $\widetilde{P}_{\mathcal{M},\text{sim}}$ | $66.47 \pm 1.11$ | $36.54 \pm 0.24$ |
| | $\widetilde{P}_{\mathcal{M}}$ | $65.39 \pm 1.05$ | $44.48 \pm 0.10$ |

Table 14: gemma-2-9b-it results on the GSM8k dataset.

|  |  | AUROC | ECE |
|---|---|---|---|
| Token Probability | $\overline{P}_{\mathcal{M}}(x_{\text{ans}})$ | $58.28 \pm 0.74$ | $49.77 \pm 0.01$ |
|  | $\overline{P}_{\mathcal{M}}(x_{\text{resp}})$ | $65.32 \pm 0.70$ | $43.32 \pm 0.03$ |
|  | $P_{\mathcal{M}}(x_{\text{ans}}|x_{\text{cot}}, x_{\text{instr}})$ | $58.06 \pm 0.70$ | $49.69 \pm 0.05$ |
|  | $P_{\mathcal{M}}(x_{\text{resp}}|x_{\text{instr}})$ | $71.01 \pm 0.55$ | $50.00 \pm 0.00$ |
| Token Entropy | $\overline{\mathcal{H}}$ | $65.58 \pm 0.70$ | $33.02 \pm 0.06$ |
|  | $\overline{\mathcal{H}}(x_{\text{ans}})$ | $58.32 \pm 0.73$ | $49.30 \pm 0.02$ |
|  | $\mathcal{H}$ | $71.43 \pm 0.53$ | - |
|  | $\mathcal{H}(x_{\text{ans}})$ | $58.06 \pm 0.70$ | - |
| Multi-Round Prompting | Verbalized Uncertainty | $52.08 \pm 0.44$ | $42.96 \pm 0.09$ |
|  | Self-Consistency | $75.12 \pm 0.47$ | $27.65 \pm 0.23$ |
| UQAC | $\overline{P}_{\mathcal{M},\text{attn}}$ | $59.48 \pm 0.71$ | $43.32 \pm 0.03$ |
|  | $\overline{P}_{\mathcal{M},\text{sim}}$ | $58.18 \pm 0.98$ | $46.79 \pm 0.07$ |
|  | $\widetilde{P}_{\mathcal{M},\text{attn}}$ | $58.96 \pm 0.67$ | $40.42 \pm 0.41$ |
|  | $\widetilde{P}_{\mathcal{M},\text{sim}}$ | $57.82 \pm 0.97$ | $28.13 \pm 0.63$ |
|  | $\widetilde{P}_{\mathcal{M}}$ | $57.97 \pm 0.60$ | $34.76 \pm 0.27$ |

Table 15: Qwen2.5-1.5B-Instruct results on the GSM8k dataset.

|  |  | AUROC | ECE |
|---|---|---|---|
| Token Probability | $\overline{P}_{\mathcal{M}}(x_{\text{ans}})$ | $57.17 \pm 0.18$ | $49.82 \pm 0.03$ |
|  | $\overline{P}_{\mathcal{M}}(x_{\text{resp}})$ | $57.29 \pm 0.89$ | $42.45 \pm 0.04$ |
|  | $P_{\mathcal{M}}(x_{\text{ans}}|x_{\text{cot}}, x_{\text{instr}})$ | $56.52 \pm 0.22$ | $49.44 \pm 0.07$ |
|  | $P_{\mathcal{M}}(x_{\text{resp}}|x_{\text{instr}})$ | $67.71 \pm 0.42$ | $50.00 \pm 0.01$ |
| Token Entropy | $\overline{\mathcal{H}}$ | $57.26 \pm 0.90$ | $30.84 \pm 0.10$ |
|  | $\overline{\mathcal{H}}(x_{\text{ans}})$ | $57.40 \pm 0.12$ | $49.53 \pm 0.07$ |
|  | $\mathcal{H}$ | $68.44 \pm 0.34$ | - |
|  | $\mathcal{H}(x_{\text{ans}})$ | $56.00 \pm 0.20$ | - |
| Multi-Round Prompting | Verbalized Uncertainty | $51.68 \pm 0.10$ | $49.77 \pm 0.01$ |
|  | Self-Consistency | $71.88 \pm 0.31$ | $33.92 \pm 0.14$ |
| UQAC | $\overline{P}_{\mathcal{M},\text{attn}}$ | $52.84 \pm 1.20$ | $42.74 \pm 0.08$ |
|  | $\overline{P}_{\mathcal{M},\text{sim}}$ | $57.13 \pm 1.42$ | $46.42 \pm 0.10$ |
|  | $\widetilde{P}_{\mathcal{M},\text{attn}}$ | $56.48 \pm 0.58$ | $43.01 \pm 0.68$ |
|  | $\widetilde{P}_{\mathcal{M},\text{sim}}$ | $56.96 \pm 1.41$ | $31.85 \pm 0.46$ |
|  | $\widetilde{P}_{\mathcal{M}}$ | $56.78 \pm 1.10$ | $36.90 \pm 0.46$ |

Table 16: Qwen2.5-3B-Instruct results on the GSM8k dataset.

|  |  | AUROC | ECE |
|---|---|---|---|
| Token Probability | $\overline{P}_{\mathcal{M}}(\boldsymbol{x}_{\text{ans}})$ | $53.54 \pm 0.95$ | $49.82 \pm 0.01$ |
|  | $\overline{P}_{\mathcal{M}}(\boldsymbol{x}_{\text{resp}})$ | $54.21 \pm 1.24$ | $44.53 \pm 0.03$ |
|  | $P_{\mathcal{M}}(\boldsymbol{x}_{\text{ans}}|\boldsymbol{x}_{\text{cot}}, \boldsymbol{x}_{\text{instr}})$ | $52.92 \pm 0.99$ | $49.58 \pm 0.03$ |
|  | $P_{\mathcal{M}}(\boldsymbol{x}_{\text{resp}}|\boldsymbol{x}_{\text{instr}})$ | $61.27 \pm 1.26$ | $49.98 \pm 0.02$ |
| Token Entropy | $\overline{\mathcal{H}}$ | $54.70 \pm 1.07$ | $36.19 \pm 0.07$ |
|  | $\overline{\mathcal{H}}(\boldsymbol{x}_{\text{ans}})$ | $49.25 \pm 1.01$ | $49.65 \pm 0.04$ |
|  | $\mathcal{H}$ | $61.88 \pm 1.11$ | - |
|  | $\mathcal{H}(\boldsymbol{x}_{\text{ans}})$ | $46.73 \pm 1.11$ | - |
| Multi-Round Prompting | Verbalized Uncertainty | $53.77 \pm 0.37$ | $49.04 \pm 0.05$ |
|  | Self-Consistency | $64.74 \pm 1.06$ | $41.17 \pm 0.35$ |
| UQAC | $\overline{P}_{\mathcal{M},\text{attn}}$ | $50.55 \pm 1.21$ | $46.80 \pm 0.07$ |
|  | $\overline{P}_{\mathcal{M},\text{sim}}$ | $53.08 \pm 0.95$ | $47.61 \pm 0.05$ |
|  | $\widetilde{P}_{\mathcal{M},\text{attn}}$ | $53.18 \pm 1.18$ | $27.35 \pm 0.89$ |
|  | $\widetilde{P}_{\mathcal{M},\text{sim}}$ | $52.91 \pm 0.93$ | $33.34 \pm 0.23$ |
|  | $\widetilde{P}_{\mathcal{M}}$ | $49.98 \pm 0.83$ | $42.39 \pm 0.26$ |

Table 17: Qwen2.5-7B-Instruct results on the GSM8k dataset.

|  |  | AUROC | ECE |
|---|---|---|---|
| Token Probability | $\overline{P}_{\mathcal{M}}(\boldsymbol{x}_{\text{ans}})$ | $76.36 \pm 0.46$ | $47.62 \pm 0.03$ |
|  | $\overline{P}_{\mathcal{M}}(\boldsymbol{x}_{\text{resp}})$ | $56.52 \pm 0.22$ | $41.55 \pm 0.02$ |
|  | $P_{\mathcal{M}}(\boldsymbol{x}_{\text{ans}}|\boldsymbol{x}_{\text{cot}}, \boldsymbol{x}_{\text{instr}})$ | $77.25 \pm 0.40$ | $41.03 \pm 0.07$ |
|  | $P_{\mathcal{M}}(\boldsymbol{x}_{\text{resp}}|\boldsymbol{x}_{\text{instr}})$ | $61.04 \pm 0.74$ | $50.00 \pm 0.00$ |
| Token Entropy | $\overline{\mathcal{H}}$ | $56.49 \pm 0.27$ | $31.02 \pm 0.10$ |
|  | $\overline{\mathcal{H}}(\boldsymbol{x}_{\text{ans}})$ | $76.10 \pm 0.49$ | $44.46 \pm 0.08$ |
|  | $\mathcal{H}$ | $61.06 \pm 0.84$ | - |
|  | $\mathcal{H}(\boldsymbol{x}_{\text{ans}})$ | $77.39 \pm 0.39$ | - |
| Multi-Round Prompting | Verbalized Uncertainty | $48.06 \pm 1.25$ | $37.65 \pm 0.98$ |
|  | Self-Consistency | $82.62 \pm 0.59$ | $11.41 \pm 0.40$ |
| UQAC | $\overline{P}_{\mathcal{M},\text{attn}}$ | $55.93 \pm 0.45$ | $41.38 \pm 0.02$ |
|  | $\overline{P}_{\mathcal{M},\text{sim}}$ | $63.58 \pm 0.71$ | $46.10 \pm 0.05$ |
|  | $\widetilde{P}_{\mathcal{M},\text{attn}}$ | $57.92 \pm 0.54$ | $43.37 \pm 0.31$ |
|  | $\widetilde{P}_{\mathcal{M},\text{sim}}$ | $66.38 \pm 0.70$ | $19.82 \pm 0.61$ |
|  | $\widetilde{P}_{\mathcal{M}}$ | $70.96 \pm 0.85$ | $25.06 \pm 0.24$ |

Table 18: DeepSeek-R1-Distill-Llama-8B results on the GSM8k dataset.

|  |  | AUROC | ECE |
|---|---|---|---|
| Token Probability | $\overline{P}_{\mathcal{M}}(x_{\text{ans}})$ | $69.59 \pm 0.85$ | $48.91 \pm 0.17$ |
|  | $\overline{P}_{\mathcal{M}}(x_{\text{resp}})$ | $66.19 \pm 1.58$ | $40.13 \pm 0.06$ |
|  | $P_{\mathcal{M}}(x_{\text{ans}}|x_{\text{cot}}, x_{\text{instr}})$ | $68.90 \pm 0.86$ | $46.71 \pm 0.26$ |
|  | $P_{\mathcal{M}}(x_{\text{resp}}|x_{\text{instr}})$ | $79.93 \pm 0.90$ | $50.00 \pm 0.00$ |
| Token Entropy | $\overline{\mathcal{H}}$ | $65.87 \pm 1.59$ | $27.73 \pm 0.09$ |
|  | $\overline{\mathcal{H}}(x_{\text{ans}})$ | $69.52 \pm 0.81$ | $48.02 \pm 0.30$ |
|  | $\mathcal{H}$ | $78.93 \pm 0.73$ | - |
|  | $\mathcal{H}(x_{\text{ans}})$ | $68.79 \pm 0.81$ | - |
| Multi-Round Prompting | Verbalized Uncertainty | $55.25 \pm 0.74$ | $44.05 \pm 0.51$ |
|  | Self-Consistency | $82.03 \pm 0.80$ | $9.77 \pm 1.04$ |
| UQAC | $\overline{P}_{\mathcal{M},\text{attn}}$ | $66.92 \pm 0.95$ | $39.64 \pm 0.10$ |
|  | $\overline{P}_{\mathcal{M},\text{sim}}$ | $65.90 \pm 1.33$ | $44.94 \pm 0.10$ |
|  | $\widetilde{P}_{\mathcal{M},\text{attn}}$ | $69.48 \pm 1.41$ | $47.48 \pm 0.49$ |
|  | $\widetilde{P}_{\mathcal{M},\text{sim}}$ | $67.22 \pm 1.19$ | $17.48 \pm 0.96$ |
|  | $\widetilde{P}_{\mathcal{M}}$ | $67.92 \pm 1.07$ | $20.65 \pm 0.51$ |

Table 19: Llama-3.2-1B-Instruct results on the MATH dataset.

|  |  | AUROC | ECE |
|---|---|---|---|
| Token Probability | $\overline{P}_{\mathcal{M}}(x_{\text{ans}})$ | $70.93 \pm 1.37$ | $47.15 \pm 0.17$ |
|  | $\overline{P}_{\mathcal{M}}(x_{\text{resp}})$ | $64.72 \pm 1.00$ | $40.57 \pm 0.12$ |
|  | $P_{\mathcal{M}}(x_{\text{ans}}|x_{\text{cot}}, x_{\text{instr}})$ | $70.82 \pm 1.46$ | $42.62 \pm 0.25$ |
|  | $P_{\mathcal{M}}(x_{\text{resp}}|x_{\text{instr}})$ | $75.37 \pm 1.50$ | $50.00 \pm 0.00$ |
| Token Entropy | $\overline{\mathcal{H}}$ | $64.56 \pm 0.99$ | $28.73 \pm 0.28$ |
|  | $\overline{\mathcal{H}}(x_{\text{ans}})$ | $70.87 \pm 1.35$ | $45.18 \pm 0.37$ |
|  | $\mathcal{H}$ | $74.59 \pm 1.59$ | - |
|  | $\mathcal{H}(x_{\text{ans}})$ | $70.79 \pm 1.45$ | - |
| Multi-Round Prompting | Verbalized Uncertainty | $60.98 \pm 0.90$ | $41.51 \pm 0.21$ |
|  | Self-Consistency | $74.04 \pm 0.55$ | $12.66 \pm 0.88$ |
| UQAC | $\overline{P}_{\mathcal{M},\text{attn}}$ | $63.62 \pm 1.13$ | $40.69 \pm 0.12$ |
|  | $\overline{P}_{\mathcal{M},\text{sim}}$ | $65.04 \pm 0.76$ | $44.75 \pm 0.06$ |
|  | $\widetilde{P}_{\mathcal{M},\text{attn}}$ | $63.95 \pm 1.61$ | $46.88 \pm 0.68$ |
|  | $\widetilde{P}_{\mathcal{M},\text{sim}}$ | $66.58 \pm 0.83$ | $20.92 \pm 0.84$ |
|  | $\widetilde{P}_{\mathcal{M}}$ | $68.40 \pm 0.89$ | $21.14 \pm 0.81$ |

Table 20: Llama-3.2-3B-Instruct results on the MATH dataset.

|  |  | AUROC | ECE |
|---|---|---|---|
| Token Probability | $\overline{P}_{\mathcal{M}}(x_{\text{ans}})$ | $77.71 \pm 1.72$ | $48.10 \pm 0.25$ |
|  | $\overline{P}_{\mathcal{M}}(x_{\text{resp}})$ | $68.62 \pm 1.80$ | $40.41 \pm 0.13$ |
|  | $P_{\mathcal{M}}(x_{\text{ans}}|x_{\text{cot}}, x_{\text{instr}})$ | $76.59 \pm 1.69$ | $44.93 \pm 0.48$ |
|  | $P_{\mathcal{M}}(x_{\text{resp}}|x_{\text{instr}})$ | $81.03 \pm 1.16$ | $50.00 \pm 0.00$ |
| Token Entropy | $\overline{\mathcal{H}}$ | $68.19 \pm 1.73$ | $27.52 \pm 0.26$ |
|  | $\overline{\mathcal{H}}(x_{\text{ans}})$ | $77.72 \pm 1.69$ | $46.83 \pm 0.48$ |
|  | $\mathcal{H}$ | $80.38 \pm 1.20$ | - |
|  | $\mathcal{H}(x_{\text{ans}})$ | $76.45 \pm 1.68$ | - |
| Multi-Round Prompting | Verbalized Uncertainty | $60.17 \pm 0.75$ | $46.44 \pm 0.33$ |
|  | Self-Consistency | $82.01 \pm 1.53$ | $14.99 \pm 0.54$ |
| UQAC | $\overline{P}_{\mathcal{M},\text{attn}}$ | $65.01 \pm 1.63$ | $39.46 \pm 0.17$ |
|  | $\overline{P}_{\mathcal{M},\text{sim}}$ | $66.53 \pm 1.24$ | $44.92 \pm 0.18$ |
|  | $\widetilde{P}_{\mathcal{M},\text{attn}}$ | $64.70 \pm 0.68$ | $45.17 \pm 0.45$ |
|  | $\widetilde{P}_{\mathcal{M},\text{sim}}$ | $67.86 \pm 1.02$ | $18.57 \pm 1.02$ |
|  | $\widetilde{P}_{\mathcal{M}}$ | $70.00 \pm 1.14$ | $21.97 \pm 1.62$ |

Table 21: Meta-Llama-3.1-8B-Instruct results on the MATH dataset.

|  |  | AUROC | ECE |
|---|---|---|---|
| Token Probability | $\overline{P}_{\mathcal{M}}(x_{\text{ans}})$ | $73.22 \pm 0.73$ | $45.91 \pm 0.18$ |
|  | $\overline{P}_{\mathcal{M}}(x_{\text{resp}})$ | $71.01 \pm 1.38$ | $37.54 \pm 0.09$ |
|  | $P_{\mathcal{M}}(x_{\text{ans}}|x_{\text{cot}}, x_{\text{instr}})$ | $72.82 \pm 0.82$ | $34.13 \pm 0.25$ |
|  | $P_{\mathcal{M}}(x_{\text{resp}}|x_{\text{instr}})$ | $78.88 \pm 0.62$ | $50.00 \pm 0.00$ |
| Token Entropy | $\overline{\mathcal{H}}$ | $71.46 \pm 1.31$ | $21.70 \pm 0.22$ |
|  | $\overline{\mathcal{H}}(x_{\text{ans}})$ | $73.33 \pm 0.75$ | $40.67 \pm 0.23$ |
|  | $\mathcal{H}$ | $77.26 \pm 0.76$ | - |
|  | $\mathcal{H}(x_{\text{ans}})$ | $72.76 \pm 0.85$ | - |
| Multi-Round Prompting | Verbalized Uncertainty | $60.26 \pm 0.65$ | $40.80 \pm 0.26$ |
|  | Self-Consistency | $77.26 \pm 0.80$ | $18.44 \pm 1.93$ |
| UQAC | $\overline{P}_{\mathcal{M},\text{attn}}$ | $72.97 \pm 1.27$ | $39.76 \pm 0.05$ |
|  | $\overline{P}_{\mathcal{M},\text{sim}}$ | $70.97 \pm 1.06$ | $43.38 \pm 0.10$ |
|  | $\widetilde{P}_{\mathcal{M},\text{attn}}$ | $74.57 \pm 1.32$ | $43.64 \pm 0.21$ |
|  | $\widetilde{P}_{\mathcal{M},\text{sim}}$ | $72.91 \pm 1.29$ | $15.74 \pm 1.48$ |
|  | $\widetilde{P}_{\mathcal{M}}$ | $74.09 \pm 1.18$ | $15.23 \pm 0.73$ |

Table 22: gemma-2-2b-it results on the MATH dataset.

|  |  | AUROC | ECE |
|---|---|---|---|
| Token Probability | $\overline{P}_{\mathcal{M}}(x_{\text{ans}})$ | $69.26 \pm 0.78$ | $48.62 \pm 0.15$ |
|  | $\overline{P}_{\mathcal{M}}(x_{\text{resp}})$ | $67.87 \pm 1.50$ | $43.17 \pm 0.05$ |
|  | $P_{\mathcal{M}}(x_{\text{ans}}|x_{\text{cot}}, x_{\text{instr}})$ | $69.22 \pm 0.72$ | $43.89 \pm 0.35$ |
|  | $P_{\mathcal{M}}(x_{\text{resp}}|x_{\text{instr}})$ | $76.38 \pm 1.18$ | $50.00 \pm 0.00$ |
| Token Entropy | $\overline{\mathcal{H}}$ | $70.38 \pm 1.38$ | $34.50 \pm 0.12$ |
|  | $\overline{\mathcal{H}}(x_{\text{ans}})$ | $69.35 \pm 0.76$ | $46.85 \pm 0.25$ |
|  | $\mathcal{H}$ | $77.39 \pm 1.10$ | - |
|  | $\mathcal{H}(x_{\text{ans}})$ | $69.33 \pm 0.72$ | - |
| Multi-Round Prompting | Verbalized Uncertainty | $57.55 \pm 0.51$ | $49.01 \pm 0.05$ |
|  | Self-Consistency | $84.51 \pm 1.13$ | $19.22 \pm 0.50$ |
| UQAC | $\overline{P}_{\mathcal{M},\text{attn}}$ | $67.12 \pm 1.43$ | $44.10 \pm 0.10$ |
|  | $\overline{P}_{\mathcal{M},\text{sim}}$ | $62.52 \pm 1.04$ | $47.07 \pm 0.11$ |
|  | $\widetilde{P}_{\mathcal{M},\text{attn}}$ | $70.35 \pm 1.68$ | $38.73 \pm 0.63$ |
|  | $\widetilde{P}_{\mathcal{M},\text{sim}}$ | $63.72 \pm 1.07$ | $24.12 \pm 1.09$ |
|  | $\widetilde{P}_{\mathcal{M}}$ | $64.21 \pm 0.88$ | $31.06 \pm 0.93$ |

Table 23: gemma-2-9b-it results on the MATH dataset.

|  |  | AUROC | ECE |
|---|---|---|---|
| Token Probability | $\overline{P}_{\mathcal{M}}(x_{\text{ans}})$ | $79.43 \pm 1.85$ | $48.69 \pm 0.11$ |
|  | $\overline{P}_{\mathcal{M}}(x_{\text{resp}})$ | $73.66 \pm 2.22$ | $44.35 \pm 0.04$ |
|  | $P_{\mathcal{M}}(x_{\text{ans}}|x_{\text{cot}}, x_{\text{instr}})$ | $79.72 \pm 1.66$ | $46.07 \pm 0.32$ |
|  | $P_{\mathcal{M}}(x_{\text{resp}}|x_{\text{instr}})$ | $82.17 \pm 1.66$ | $50.00 \pm 0.00$ |
| Token Entropy | $\overline{\mathcal{H}}$ | $73.73 \pm 2.22$ | $35.76 \pm 0.09$ |
|  | $\overline{\mathcal{H}}(x_{\text{ans}})$ | $79.44 \pm 1.85$ | $46.53 \pm 0.37$ |
|  | $\mathcal{H}$ | $82.14 \pm 1.63$ | - |
|  | $\mathcal{H}(x_{\text{ans}})$ | $79.75 \pm 1.64$ | - |
| Multi-Round Prompting | Verbalized Uncertainty | $55.09 \pm 0.90$ | $44.61 \pm 0.32$ |
|  | Self-Consistency | $78.30 \pm 0.53$ | $14.58 \pm 0.34$ |
| UQAC | $\overline{P}_{\mathcal{M},\text{attn}}$ | $67.87 \pm 1.14$ | $44.10 \pm 0.06$ |
|  | $\overline{P}_{\mathcal{M},\text{sim}}$ | $69.33 \pm 2.06$ | $47.25 \pm 0.10$ |
|  | $\widetilde{P}_{\mathcal{M},\text{attn}}$ | $68.17 \pm 0.84$ | $39.55 \pm 0.46$ |
|  | $\widetilde{P}_{\mathcal{M},\text{sim}}$ | $70.17 \pm 2.09$ | $22.90 \pm 0.90$ |
|  | $\widetilde{P}_{\mathcal{M}}$ | $71.71 \pm 2.01$ | $31.45 \pm 0.72$ |

Table 24: Qwen2.5-1.5B-Instruct results on the MATH dataset.

| | | AUROC | ECE |
|---|---|---|---|
| Token Probability | $\overline{P}_{\mathcal{M}}(x_{\text{ans}})$ | $79.03 \pm 0.83$ | $49.34 \pm 0.08$ |
| | $\overline{P}_{\mathcal{M}}(x_{\text{resp}})$ | $71.05 \pm 1.11$ | $44.50 \pm 0.10$ |
| | $P_{\mathcal{M}}(x_{\text{ans}}|x_{\text{cot}}, x_{\text{instr}})$ | $78.94 \pm 0.89$ | $47.61 \pm 0.10$ |
| | $P_{\mathcal{M}}(x_{\text{resp}}|x_{\text{instr}})$ | $79.30 \pm 0.72$ | $50.00 \pm 0.00$ |
| Token Entropy | $\overline{\mathcal{H}}$ | $70.92 \pm 1.14$ | $36.13 \pm 0.24$ |
| | $\overline{\mathcal{H}}(x_{\text{ans}})$ | $79.21 \pm 0.77$ | $48.08 \pm 0.19$ |
| | $\mathcal{H}$ | $79.53 \pm 0.67$ | - |
| | $\mathcal{H}(x_{\text{ans}})$ | $79.14 \pm 0.87$ | - |
| Multi-Round Prompting | Verbalized Uncertainty | $52.46 \pm 0.34$ | $49.56 \pm 0.04$ |
| | Self-Consistency | $78.26 \pm 1.77$ | $20.61 \pm 0.43$ |
| UQAC | $\overline{P}_{\mathcal{M},\text{attn}}$ | $67.15 \pm 0.95$ | $45.03 \pm 0.10$ |
| | $\overline{P}_{\mathcal{M},\text{sim}}$ | $66.73 \pm 1.31$ | $48.29 \pm 0.05$ |
| | $\widetilde{P}_{\mathcal{M},\text{attn}}$ | $68.94 \pm 0.67$ | $37.92 \pm 0.37$ |
| | $\widetilde{P}_{\mathcal{M},\text{sim}}$ | $67.37 \pm 1.45$ | $31.73 \pm 0.44$ |
| | $\widetilde{P}_{\mathcal{M}}$ | $69.92 \pm 1.24$ | $38.80 \pm 0.63$ |

Table 25: Qwen2.5-3B-Instruct results on the MATH dataset.

| | | AUROC | ECE |
|---|---|---|---|
| Token Probability | $\overline{P}_{\mathcal{M}}(x_{\text{ans}})$ | $77.65 \pm 1.05$ | $49.26 \pm 0.10$ |
| | $\overline{P}_{\mathcal{M}}(x_{\text{resp}})$ | $68.63 \pm 1.31$ | $45.43 \pm 0.05$ |
| | $P_{\mathcal{M}}(x_{\text{ans}}|x_{\text{cot}}, x_{\text{instr}})$ | $77.97 \pm 1.16$ | $47.78 \pm 0.15$ |
| | $P_{\mathcal{M}}(x_{\text{resp}}|x_{\text{instr}})$ | $79.42 \pm 0.57$ | $50.00 \pm 0.00$ |
| Token Entropy | $\overline{\mathcal{H}}$ | $68.82 \pm 1.32$ | $38.54 \pm 0.12$ |
| | $\overline{\mathcal{H}}(x_{\text{ans}})$ | $79.34 \pm 0.95$ | $48.09 \pm 0.19$ |
| | $\mathcal{H}$ | $79.53 \pm 0.54$ | - |
| | $\mathcal{H}(x_{\text{ans}})$ | $79.12 \pm 1.08$ | - |
| Multi-Round Prompting | Verbalized Uncertainty | $61.96 \pm 0.46$ | $47.34 \pm 0.12$ |
| | Self-Consistency | $74.83 \pm 0.42$ | $25.52 \pm 0.47$ |
| UQAC | $\overline{P}_{\mathcal{M},\text{attn}}$ | $57.85 \pm 0.59$ | $47.46 \pm 0.13$ |
| | $\overline{P}_{\mathcal{M},\text{sim}}$ | $58.33 \pm 0.94$ | $48.04 \pm 0.12$ |
| | $\widetilde{P}_{\mathcal{M},\text{attn}}$ | $59.28 \pm 0.66$ | $25.96 \pm 0.94$ |
| | $\widetilde{P}_{\mathcal{M},\text{sim}}$ | $59.42 \pm 0.91$ | $34.36 \pm 0.64$ |
| | $\widetilde{P}_{\mathcal{M}}$ | $62.58 \pm 1.19$ | $40.80 \pm 0.43$ |

Table 26: Qwen2.5-7B-Instruct results on the MATH dataset.

|  |  | AUROC | ECE |
|---|---|---|---|
| Token Probability | $\overline{P}_{\mathcal{M}}(x_{\text{ans}})$ | $82.39 \pm 1.01$ | $48.62 \pm 0.00$ |
|  | $\overline{P}_{\mathcal{M}}(x_{\text{resp}})$ | $37.31 \pm 2.22$ | $39.63 \pm 0.10$ |
|  | $P_{\mathcal{M}}(x_{\text{ans}}|x_{\text{cot}}, x_{\text{instr}})$ | $81.40 \pm 1.12$ | $44.91 \pm 0.15$ |
|  | $P_{\mathcal{M}}(x_{\text{resp}}|x_{\text{instr}})$ | $33.70 \pm 1.31$ | $50.00 \pm 0.00$ |
| Token Entropy | $\overline{\mathcal{H}}$ | $36.18 \pm 2.30$ | $26.65 \pm 0.35$ |
|  | $\overline{\mathcal{H}}(x_{\text{ans}})$ | $82.22 \pm 1.06$ | $46.54 \pm 0.03$ |
|  | $\mathcal{H}$ | $30.75 \pm 1.81$ | - |
|  | $\mathcal{H}(x_{\text{ans}})$ | $81.28 \pm 1.18$ | - |
| Multi-Round Prompting | Verbalized Uncertainty | $47.36 \pm 3.65$ | $36.77 \pm 2.41$ |
|  | Self-Consistency | $65.51 \pm 1.39$ | $22.00 \pm 1.70$ |
| UQAC | $\overline{P}_{\mathcal{M},\text{attn}}$ | $56.11 \pm 1.24$ | $40.43 \pm 0.11$ |
|  | $\overline{P}_{\mathcal{M},\text{sim}}$ | $67.82 \pm 1.06$ | $46.39 \pm 0.07$ |
|  | $\widetilde{P}_{\mathcal{M},\text{attn}}$ | $52.44 \pm 1.61$ | $44.73 \pm 0.78$ |
|  | $\widetilde{P}_{\mathcal{M},\text{sim}}$ | $69.00 \pm 0.99$ | $20.92 \pm 1.65$ |
|  | $\widetilde{P}_{\mathcal{M}}$ | $69.94 \pm 1.08$ | $27.94 \pm 0.59$ |

Table 27: DeepSeek-R1-Distill-Llama-8B results on the MATH dataset.

|  |  | AUROC | ECE |
|---|---|---|---|
| Token Probability | $\overline{P}_{\mathcal{M}}(x_{\text{ans}})$ | $53.05 \pm 0.69$ | $36.02 \pm 1.03$ |
|  | $\overline{P}_{\mathcal{M}}(x_{\text{resp}})$ | $58.98 \pm 2.07$ | $40.69 \pm 0.19$ |
|  | $P_{\mathcal{M}}(x_{\text{ans}}|x_{\text{cot}}, x_{\text{instr}})$ | $58.57 \pm 0.41$ | $25.95 \pm 1.13$ |
|  | $P_{\mathcal{M}}(x_{\text{resp}}|x_{\text{instr}})$ | $56.71 \pm 1.36$ | $33.73 \pm 0.99$ |
| Token Entropy | $\overline{\mathcal{H}}$ | $62.18 \pm 1.79$ | $31.12 \pm 0.46$ |
|  | $\overline{\mathcal{H}}(x_{\text{ans}})$ | $52.12 \pm 0.96$ | $29.82 \pm 1.10$ |
|  | $\mathcal{H}$ | $57.68 \pm 1.37$ | - |
|  | $\mathcal{H}(x_{\text{ans}})$ | $61.60 \pm 0.75$ | - |
| Multi-Round Prompting | Verbalized Uncertainty | $66.67 \pm 0.00$ | $46.67 \pm 0.00$ |
|  | Self-Consistency | $71.37 \pm 1.44$ | $22.10 \pm 0.52$ |
| UQAC | $\overline{P}_{\mathcal{M},\text{attn}}$ | $56.81 \pm 2.01$ | $38.76 \pm 0.22$ |
|  | $\overline{P}_{\mathcal{M},\text{sim}}$ | $54.66 \pm 2.13$ | $36.56 \pm 0.57$ |
|  | $\widetilde{P}_{\mathcal{M},\text{attn}}$ | $60.97 \pm 1.40$ | $24.97 \pm 1.09$ |
|  | $\widetilde{P}_{\mathcal{M},\text{sim}}$ | $59.65 \pm 1.63$ | $24.11 \pm 0.94$ |
|  | $\widetilde{P}_{\mathcal{M}}$ | $61.91 \pm 1.21$ | $21.05 \pm 0.69$ |

Table 28: Llama-3.2-1B-Instruct results on the BBH dataset.

|  |  | AUROC | ECE |
|---|---|---|---|
| Token Probability | $\overline{P}_{\mathcal{M}}(\boldsymbol{x}_{\text{ans}})$ | $64.01 \pm 1.24$ | $40.60 \pm 0.32$ |
|  | $\overline{P}_{\mathcal{M}}(\boldsymbol{x}_{\text{resp}})$ | $63.39 \pm 0.47$ | $44.14 \pm 0.14$ |
|  | $P_{\mathcal{M}}(\boldsymbol{x}_{\text{ans}}|\boldsymbol{x}_{\text{cot}}, \boldsymbol{x}_{\text{instr}})$ | $64.22 \pm 1.08$ | $27.21 \pm 0.82$ |
|  | $P_{\mathcal{M}}(\boldsymbol{x}_{\text{resp}}|\boldsymbol{x}_{\text{instr}})$ | $66.29 \pm 2.03$ | $45.21 \pm 0.49$ |
| Token Entropy | $\overline{\mathcal{H}}$ | $61.60 \pm 0.49$ | $36.92 \pm 0.26$ |
|  | $\overline{\mathcal{H}}(\boldsymbol{x}_{\text{ans}})$ | $64.23 \pm 1.16$ | $31.08 \pm 0.39$ |
|  | $\mathcal{H}$ | $63.66 \pm 1.99$ | - |
|  | $\mathcal{H}(\boldsymbol{x}_{\text{ans}})$ | $64.48 \pm 0.66$ | - |
| Multi-Round Prompting | Verbalized Uncertainty | $59.56 \pm 2.39$ | $29.45 \pm 0.49$ |
|  | Self-Consistency | $87.90 \pm 1.67$ | $27.41 \pm 1.04$ |
| UQAC | $\overline{P}_{\mathcal{M},\text{attn}}$ | $66.16 \pm 1.15$ | $42.98 \pm 0.17$ |
|  | $\overline{P}_{\mathcal{M},\text{sim}}$ | $65.93 \pm 0.91$ | $42.65 \pm 0.31$ |
|  | $\widetilde{P}_{\mathcal{M},\text{attn}}$ | $65.81 \pm 1.17$ | $22.58 \pm 0.80$ |
|  | $\widetilde{P}_{\mathcal{M},\text{sim}}$ | $66.17 \pm 0.54$ | $18.94 \pm 0.66$ |
|  | $\widetilde{P}_{\mathcal{M}}$ | $65.86 \pm 0.36$ | $19.63 \pm 0.69$ |

Table 29: Llama-3.2-3B-Instruct results on the BBH dataset.

|  |  | AUROC | ECE |
|---|---|---|---|
| Token Probability | $\overline{P}_{\mathcal{M}}(\boldsymbol{x}_{\text{ans}})$ | $61.51 \pm 1.04$ | $46.34 \pm 0.27$ |
|  | $\overline{P}_{\mathcal{M}}(\boldsymbol{x}_{\text{resp}})$ | $66.05 \pm 1.04$ | $43.92 \pm 0.05$ |
|  | $P_{\mathcal{M}}(\boldsymbol{x}_{\text{ans}}|\boldsymbol{x}_{\text{cot}}, \boldsymbol{x}_{\text{instr}})$ | $61.77 \pm 1.02$ | $42.12 \pm 0.41$ |
|  | $P_{\mathcal{M}}(\boldsymbol{x}_{\text{resp}}|\boldsymbol{x}_{\text{instr}})$ | $73.38 \pm 0.83$ | $46.61 \pm 0.23$ |
| Token Entropy | $\overline{\mathcal{H}}$ | $63.96 \pm 1.11$ | $34.84 \pm 0.13$ |
|  | $\overline{\mathcal{H}}(\boldsymbol{x}_{\text{ans}})$ | $61.54 \pm 1.05$ | $42.15 \pm 0.43$ |
|  | $\mathcal{H}$ | $71.76 \pm 0.98$ | - |
|  | $\mathcal{H}(\boldsymbol{x}_{\text{ans}})$ | $61.97 \pm 1.03$ | - |
| Multi-Round Prompting | Verbalized Uncertainty | $58.82 \pm 1.27$ | $30.91 \pm 0.53$ |
|  | Self-Consistency | $84.08 \pm 0.97$ | $35.61 \pm 0.56$ |
| UQAC | $\overline{P}_{\mathcal{M},\text{attn}}$ | $67.65 \pm 0.61$ | $43.65 \pm 0.06$ |
|  | $\overline{P}_{\mathcal{M},\text{sim}}$ | $67.79 \pm 0.37$ | $44.15 \pm 0.07$ |
|  | $\widetilde{P}_{\mathcal{M},\text{attn}}$ | $68.79 \pm 1.14$ | $21.15 \pm 1.17$ |
|  | $\widetilde{P}_{\mathcal{M},\text{sim}}$ | $68.58 \pm 0.33$ | $19.79 \pm 0.44$ |
|  | $\widetilde{P}_{\mathcal{M}}$ | $69.75 \pm 0.71$ | $23.04 \pm 0.56$ |

Table 30: Meta-Llama-3.1-8B-Instruct results on the BBH dataset.

|  |  | AUROC | ECE |
|---|---|---|---|
| Token Probability | $\overline{P}_{\mathcal{M}}(x_{\text{ans}})$ | $59.94 \pm 1.00$ | $46.09 \pm 0.22$ |
|  | $\overline{P}_{\mathcal{M}}(x_{\text{resp}})$ | $52.70 \pm 1.19$ | $42.80 \pm 0.33$ |
|  | $P_{\mathcal{M}}(x_{\text{ans}}|x_{\text{cot}}, x_{\text{instr}})$ | $61.29 \pm 0.89$ | $39.62 \pm 0.75$ |
|  | $P_{\mathcal{M}}(x_{\text{resp}}|x_{\text{instr}})$ | $59.84 \pm 1.34$ | $50.04 \pm 0.06$ |
| Token Entropy | $\overline{\mathcal{H}}$ | $53.84 \pm 0.80$ | $36.99 \pm 0.49$ |
|  | $\overline{\mathcal{H}}(x_{\text{ans}})$ | $59.91 \pm 0.88$ | $40.89 \pm 0.40$ |
|  | $\mathcal{H}$ | $59.85 \pm 1.49$ | - |
|  | $\mathcal{H}(x_{\text{ans}})$ | $61.84 \pm 0.70$ | - |
| Multi-Round Prompting | Verbalized Uncertainty | $53.77 \pm 0.83$ | $37.84 \pm 0.07$ |
|  | Self-Consistency | $82.82 \pm 0.89$ | $29.16 \pm 1.05$ |
| UQAC | $\overline{P}_{\mathcal{M},\text{attn}}$ | $58.62 \pm 0.76$ | $43.41 \pm 0.21$ |
|  | $\overline{P}_{\mathcal{M},\text{sim}}$ | $58.05 \pm 0.93$ | $44.05 \pm 0.15$ |
|  | $\widetilde{P}_{\mathcal{M},\text{attn}}$ | $62.12 \pm 1.41$ | $25.24 \pm 0.42$ |
|  | $\widetilde{P}_{\mathcal{M},\text{sim}}$ | $59.94 \pm 1.20$ | $27.45 \pm 1.20$ |
|  | $\widetilde{P}_{\mathcal{M}}$ | $60.78 \pm 1.65$ | $26.01 \pm 1.46$ |

Table 31: gemma-2-2b-it results on the BBH dataset.

|  |  | AUROC | ECE |
|---|---|---|---|
| Token Probability | $\overline{P}_{\mathcal{M}}(x_{\text{ans}})$ | $72.17 \pm 1.37$ | $46.10 \pm 0.32$ |
|  | $\overline{P}_{\mathcal{M}}(x_{\text{resp}})$ | $59.17 \pm 1.49$ | $43.57 \pm 0.05$ |
|  | $P_{\mathcal{M}}(x_{\text{ans}}|x_{\text{cot}}, x_{\text{instr}})$ | $73.26 \pm 1.39$ | $40.54 \pm 0.13$ |
|  | $P_{\mathcal{M}}(x_{\text{resp}}|x_{\text{instr}})$ | $63.81 \pm 1.72$ | $49.53 \pm 0.04$ |
| Token Entropy | $\overline{\mathcal{H}}$ | $60.96 \pm 1.61$ | $35.69 \pm 0.13$ |
|  | $\overline{\mathcal{H}}(x_{\text{ans}})$ | $72.08 \pm 1.38$ | $42.18 \pm 0.50$ |
|  | $\mathcal{H}$ | $66.54 \pm 1.61$ | - |
|  | $\mathcal{H}(x_{\text{ans}})$ | $73.44 \pm 1.39$ | - |
| Multi-Round Prompting | Verbalized Uncertainty | $61.37 \pm 0.58$ | $45.23 \pm 0.14$ |
|  | Self-Consistency | $77.89 \pm 0.98$ | $34.13 \pm 0.20$ |
| UQAC | $\overline{P}_{\mathcal{M},\text{attn}}$ | $67.20 \pm 1.30$ | $44.88 \pm 0.08$ |
|  | $\overline{P}_{\mathcal{M},\text{sim}}$ | $71.03 \pm 1.28$ | $45.43 \pm 0.09$ |
|  | $\widetilde{P}_{\mathcal{M},\text{attn}}$ | $68.17 \pm 1.33$ | $22.51 \pm 1.26$ |
|  | $\widetilde{P}_{\mathcal{M},\text{sim}}$ | $72.46 \pm 1.25$ | $19.99 \pm 0.82$ |
|  | $\widetilde{P}_{\mathcal{M}}$ | $72.16 \pm 1.56$ | $24.45 \pm 0.82$ |

Table 32: gemma-2-9b-it results on the BBH dataset.

|  |  | AUROC | ECE |
|---|---|---|---|
| Token Probability | $\overline{P}_{\mathcal{M}}(x_{\text{ans}})$ | $70.39 \pm 0.91$ | $46.39 \pm 0.22$ |
|  | $\overline{P}_{\mathcal{M}}(x_{\text{resp}})$ | $57.94 \pm 0.87$ | $43.11 \pm 0.20$ |
|  | $P_{\mathcal{M}}(x_{\text{ans}}|x_{\text{cot}}, x_{\text{instr}})$ | $70.39 \pm 1.04$ | $42.32 \pm 0.32$ |
|  | $P_{\mathcal{M}}(x_{\text{resp}}|x_{\text{instr}})$ | $62.49 \pm 1.16$ | $46.93 \pm 0.29$ |
| Token Entropy | $\overline{\mathcal{H}}$ | $57.72 \pm 0.88$ | $33.39 \pm 0.34$ |
|  | $\overline{\mathcal{H}}(x_{\text{ans}})$ | $70.31 \pm 0.91$ | $40.41 \pm 0.19$ |
|  | $\mathcal{H}$ | $62.74 \pm 1.22$ | - |
|  | $\mathcal{H}(x_{\text{ans}})$ | $70.54 \pm 1.06$ | - |
| Multi-Round Prompting | Verbalized Uncertainty | $50.47 \pm 1.95$ | $38.19 \pm 0.81$ |
|  | Self-Consistency | $74.05 \pm 0.65$ | $32.22 \pm 0.51$ |
| UQAC | $\overline{P}_{\mathcal{M},\text{attn}}$ | $64.12 \pm 1.39$ | $43.41 \pm 0.17$ |
|  | $\overline{P}_{\mathcal{M},\text{sim}}$ | $62.08 \pm 1.40$ | $44.14 \pm 0.17$ |
|  | $\widetilde{P}_{\mathcal{M},\text{attn}}$ | $64.53 \pm 1.20$ | $21.12 \pm 0.94$ |
|  | $\widetilde{P}_{\mathcal{M},\text{sim}}$ | $63.51 \pm 1.53$ | $19.04 \pm 1.70$ |
|  | $\widetilde{P}_{\mathcal{M}}$ | $65.90 \pm 1.38$ | $21.35 \pm 1.37$ |

Table 33: Qwen2.5-1.5B-Instruct results on the BBH dataset.

|  |  | AUROC | ECE |
|---|---|---|---|
| Token Probability | $\overline{P}_{\mathcal{M}}(x_{\text{ans}})$ | $68.61 \pm 0.60$ | $46.75 \pm 0.16$ |
|  | $\overline{P}_{\mathcal{M}}(x_{\text{resp}})$ | $53.15 \pm 2.47$ | $39.20 \pm 0.22$ |
|  | $P_{\mathcal{M}}(x_{\text{ans}}|x_{\text{cot}}, x_{\text{instr}})$ | $69.10 \pm 0.57$ | $42.10 \pm 0.18$ |
|  | $P_{\mathcal{M}}(x_{\text{resp}}|x_{\text{instr}})$ | $58.17 \pm 2.18$ | $49.10 \pm 0.05$ |
| Token Entropy | $\overline{\mathcal{H}}$ | $52.67 \pm 2.47$ | $25.96 \pm 0.87$ |
|  | $\overline{\mathcal{H}}(x_{\text{ans}})$ | $68.67 \pm 0.66$ | $42.47 \pm 0.28$ |
|  | $\mathcal{H}$ | $58.46 \pm 2.07$ | - |
|  | $\mathcal{H}(x_{\text{ans}})$ | $69.37 \pm 0.60$ | - |
| Multi-Round Prompting | Verbalized Uncertainty | $53.68 \pm 1.23$ | $46.94 \pm 0.23$ |
|  | Self-Consistency | $81.87 \pm 1.20$ | $33.08 \pm 0.89$ |
| UQAC | $\overline{P}_{\mathcal{M},\text{attn}}$ | $58.28 \pm 2.10$ | $41.15 \pm 0.25$ |
|  | $\overline{P}_{\mathcal{M},\text{sim}}$ | $61.23 \pm 1.84$ | $43.21 \pm 0.20$ |
|  | $\widetilde{P}_{\mathcal{M},\text{attn}}$ | $59.01 \pm 1.78$ | $28.70 \pm 1.57$ |
|  | $\widetilde{P}_{\mathcal{M},\text{sim}}$ | $62.06 \pm 1.94$ | $23.29 \pm 1.74$ |
|  | $\widetilde{P}_{\mathcal{M}}$ | $65.82 \pm 2.09$ | $24.61 \pm 0.94$ |

Table 34: Qwen2.5-3B-Instruct results on the BBH dataset.

|  |  | AUROC | ECE |
|---|---|---|---|
| Token Probability | $\overline{P}_{\mathcal{M}}(x_{\text{ans}})$ | $70.11 \pm 0.14$ | $48.29 \pm 0.21$ |
|  | $\overline{P}_{\mathcal{M}}(x_{\text{resp}})$ | $63.18 \pm 1.15$ | $43.02 \pm 0.17$ |
|  | $P_{\mathcal{M}}(x_{\text{ans}}|x_{\text{cot}}, x_{\text{instr}})$ | $70.60 \pm 0.20$ | $44.73 \pm 0.18$ |
|  | $P_{\mathcal{M}}(x_{\text{resp}}|x_{\text{instr}})$ | $68.21 \pm 1.27$ | $40.91 \pm 0.69$ |
| Token Entropy | $\overline{\mathcal{H}}$ | $62.42 \pm 1.05$ | $32.88 \pm 0.35$ |
|  | $\overline{\mathcal{H}}(x_{\text{ans}})$ | $69.92 \pm 0.14$ | $45.44 \pm 0.37$ |
|  | $\mathcal{H}$ | $67.78 \pm 1.25$ | - |
|  | $\mathcal{H}(x_{\text{ans}})$ | $70.61 \pm 0.21$ | - |
| Multi-Round Prompting | Verbalized Uncertainty | $61.12 \pm 1.06$ | $42.73 \pm 0.18$ |
|  | Self-Consistency | $75.88 \pm 0.60$ | $39.17 \pm 0.53$ |
| UQAC | $\overline{P}_{\mathcal{M},\text{attn}}$ | $65.00 \pm 1.12$ | $44.97 \pm 0.19$ |
|  | $\overline{P}_{\mathcal{M},\text{sim}}$ | $66.28 \pm 1.04$ | $45.51 \pm 0.20$ |
|  | $\widetilde{P}_{\mathcal{M},\text{attn}}$ | $67.59 \pm 0.93$ | $20.65 \pm 0.72$ |
|  | $\widetilde{P}_{\mathcal{M},\text{sim}}$ | $68.22 \pm 1.07$ | $24.28 \pm 0.69$ |
|  | $\widetilde{P}_{\mathcal{M}}$ | $71.39 \pm 0.93$ | $33.48 \pm 0.50$ |

Table 35: Qwen2.5-7B-Instruct results on the BBH dataset.

|  |  | AUROC | ECE |
|---|---|---|---|
| Token Probability | $\overline{P}_{\mathcal{M}}(x_{\text{ans}})$ | $64.72 \pm 0.88$ | $48.12 \pm 0.11$ |
|  | $\overline{P}_{\mathcal{M}}(x_{\text{resp}})$ | $46.56 \pm 0.33$ | $35.82 \pm 0.13$ |
|  | $P_{\mathcal{M}}(x_{\text{ans}}|x_{\text{cot}}, x_{\text{instr}})$ | $66.02 \pm 0.80$ | $45.11 \pm 0.20$ |
|  | $P_{\mathcal{M}}(x_{\text{resp}}|x_{\text{instr}})$ | $41.66 \pm 0.42$ | $50.00 \pm 0.00$ |
| Token Entropy | $\overline{\mathcal{H}}$ | $46.42 \pm 0.31$ | $19.97 \pm 0.49$ |
|  | $\overline{\mathcal{H}}(x_{\text{ans}})$ | $63.81 \pm 0.94$ | $45.11 \pm 0.17$ |
|  | $\mathcal{H}$ | $41.62 \pm 0.50$ | - |
|  | $\mathcal{H}(x_{\text{ans}})$ | $65.76 \pm 0.81$ | - |
| Multi-Round Prompting | Verbalized Uncertainty | $64.20 \pm 1.41$ | $35.54 \pm 0.21$ |
|  | Self-Consistency | $74.27 \pm 1.41$ | $20.63 \pm 0.84$ |
| UQAC | $\overline{P}_{\mathcal{M},\text{attn}}$ | $49.70 \pm 1.35$ | $39.82 \pm 0.10$ |
|  | $\overline{P}_{\mathcal{M},\text{sim}}$ | $45.75 \pm 0.87$ | $42.87 \pm 0.13$ |
|  | $\widetilde{P}_{\mathcal{M},\text{attn}}$ | $53.17 \pm 1.30$ | $42.15 \pm 0.98$ |
|  | $\widetilde{P}_{\mathcal{M},\text{sim}}$ | $48.97 \pm 0.62$ | $34.50 \pm 0.28$ |
|  | $\widetilde{P}_{\mathcal{M}}$ | $48.89 \pm 0.76$ | $27.62 \pm 0.91$ |

Table 36: DeepSeek-R1-Distill-Llama-8B results on the BBH dataset.

