# OpenReview forum: "Language Model Uncertainty Quantification with Attention Chain"
_colmweb.org/COLM/2025/Conference — COLM 2025_

### Official Review · Reviewer_ZMZs · 2025-05-07

**Rating:** 4
**Confidence:** 4
**Ethics Flag:** 1

**Summary:**

This paper studies the uncertainty quantification (UQ) research problem in LLMs.  The authors propose UQAC, a confidence calibration method that iteratively constructs an attention chain of "important tokens" to the final answer via backtracking. The experimental results prove the effectiveness of UQAC.

**Questions To Authors:**

N/A

**Reasons To Accept:**

1. The research problem, LLMs' uncertainty quantification, is an interesting and important direction. This paper makes a concrete step to make some progress in this field.
2. The paper is well-written and easy to follow.
3. The proposed method UQAC, is well-formulated, and its effectiveness is proven on several benchmarks.

**Reasons To Reject:**

1. The key foundation of the proposed method is based on the attention weights distribution. This is not a very reasonable intuition to me, since there is a long debate about the reliability of the attention weights. This paper also lacks corresponding proof or experimental results to prove this fundamental assumption.
2. The authors argue the scalability of the UQAC by saying that it can handle responses of arbitrary length by isolating the most influential tokens. But I don't think the scalability should be measured in terms of handling responses of varying lengths. The method uses repetitive backtracking, similarity filtering, and probability thresholding to get the uncertainty quantification, which adds multiple layers of complexity. Compared to other methods that directly train LLMs to generate confidence scores, the proposed method is not that scalable.
3. The experimental results are weak. Only some variants (ablations) and weak baselines are considered for comparison. Maybe the authors should include more advanced LLMs calibration methods for comparison [1,2]. In addition, the proposed UQAC lags behind the self-consistency uncertainty quantification method.

[1] R-Tuning: Instructing Large Language Models to Say `I Don't Know'. Hanning Zhang et al

[2] SaySelf: Teaching LLMs to Express Confidence with Self-Reflective Rationales. Tianyang Xu et al

---

> ### Author Response · Authors · 2025-05-31
> **Response to Reviewer ZMZs**
>
> We appreciate Reviewer ZMZs's feedback. We believe there might have been some misunderstandings, which we address below for clarity:
>
> - **Intuition behind Attention Weights**: Attention weights intuitively represent the contributions of previous tokens toward predicting subsequent tokens, as discussed in Lines 49--55 and 143--145 of our manuscript. The efficacy of attention mechanisms has been validated since the very beginning of their proposal [1,2,3], both theoretically and empirically. Hence, we chose not to reiterate these theoretical foundations extensively within our paper.
> - **Scalability**: We notice a potential divergence in the interpretation of "scalability". Commonly, scalability refers to a system's ability to maintain performance when facing increased input complexity or volume (e.g., longer sequences or varied input topics), rather than algorithmic simplicity. In this context, topic-specific fine-tuning is inherently unscalable, which we want to avoid.
> - **Weak Results**: As clarified in Lines 44--46 and 104--113, fine-tuning methods necessitate task-specific labeled datasets, which are typically impractical to obtain in production environments or by end-users. Our study intentionally addresses UQ from a perspective distinct from fine-tuning. Nevertheless, we remain open to recommendations for alternative strong baselines within the scope of our work.
>
> [1] Bahdanau, Dzmitry, Kyunghyun Cho, and Yoshua Bengio. "Neural machine translation by jointly learning to align and translate." arXiv preprint arXiv:1409.0473 (2014).
> [2] Luong, Minh-Thang, Hieu Pham, and Christopher D. Manning. "Effective approaches to attention-based neural machine translation." arXiv preprint arXiv:1508.04025 (2015).
> [3] Vaswani, Ashish, et al. "Attention is all you need." Advances in neural information processing systems 30 (2017).

---

> > ### Comment · Reviewer_ZMZs · 2025-06-05
> >
> > Thanks, but the repsonse doesn't address my concern since (1) as I said, the use of attention socres is still in debate, and there is still work proving its ineffectiveness (in their experiemtns), (2) real-world deployment also requires easy-to-scale pipelines, instead of many specialied components. (3) I agree it's difficult to obtain from rela-world users. But there are still many labeled data from other domains can be utilized.

---

> > > ### Author Response · Authors · 2025-06-06
> > > **To AC**
> > >
> > > Dear Area Chair,
> > >
> > > We appreciate the reviewer's spending time leaving comments in this channel, but we found the quality of these reviews questionable.
> > >
> > > - The reviews randomly throw out the conclusion "attention is unreliable" without justification or specifying how it will impact the theoretical soundness of our paper.
> > > - The reviews totally misinterpret the well-established concept of "a scalable method".
> > > - The reviews insist on comparing apples to oranges.
> > >
> > > We welcome questions and challenges to our work, but it is hard to reach a consensus if the reviews are not aligned with our paper. We'd be grateful if these reviews are reasonably weighted while an overall assessment is composed.
> > >
> > > Best regards,
> > > Authors

---

### Official Review · Reviewer_4Zzx · 2025-05-10

**Rating:** 6
**Confidence:** 3
**Ethics Flag:** 1

**Summary:**

This paper proposes UQAC (uncertainty quantification with attention chain) for estimating the predictive uncertainty of LLMs in tasks that involve multi-step reasoning. Traditional uncertainty quantification methods suffer from overconfidence or poor scalability when applied to long, complex outputs. UQA addresses by tracing an attention chain via attention-based backtracking from the final answer tokens. Authors also propose similarity-based filtering and probability thresholding to enable efficient approximation of the models' answer confidence without out retraining, sampling or external models. Experiments are conducted on GSM8L, MATH, and BBH using various versions of LLM (from 1B to 9B), and the results show that UQAC achieves state-of-the-art calibration with low computational cost, outperforming baselines such as self-consistency and predictive entropy.

**Reasons To Accept:**

1. This paper is well-written and the motivation is easy to follow.
2. The method is novel. The attention backtracking mechanism is both intuitive and well-justified.
3. This method is efficient and applicable.
4. The experimental results (especially the ECE) mostly prove the efficiency of this method.

**Reasons To Reject:**

1. Lack of ablation study. Several core hyperparameters are manually chosen without any ablation study, including the number of attention heads (K) and the attention threshold ($\theta$). It would be better to add more ablation studies to see how the experimental results change with the hyperparameters.
2. The performance is not consistently higher than other baselines in Table 1. Although the authors give comprehensive explanations on this, the worse performance of this method compared to other baselines in some tasks will hinder the application of this method.

---

> ### Author Response · Authors · 2025-05-31
> **Response to Reviewer 4Zzx**
>
> We sincerely thank Reviewer 4Zzx for the supportive evaluation and insightful feedback. Below, we directly address the points raised:
>
> - **Ablation Study**: Our initial goal was to establish a universal set of hyperparameters that can be broadly applicable, thus we limited detailed exploration of alternative parameter values. Recognizing the importance of your suggestion, we have conducted additional studies. Please refer to the Tables included in our [response to Reviewer 5EYn](https://openreview.net/forum?id=QTrW2HWNXe&noteId=BkoDeNdPST) above. In short, it is possible to see (much) better scores with hyper-parameters tuned for models and datasets, but the parameter set we provide in the paper is generally stable across different setups. We'll include the complete results and a more detailed discussion in the revised article.
>
> - **Consistency of Performance**: We acknowledge the concern regarding performance consistency across different tasks, as highlighted in Table 1. This limitation was previously noted in the Limitation section of our paper (Lines 757--762). We are actively investigating enhancements to UQAC that will yield more stable and consistent improvements across diverse scenarios.
>
> We appreciate your constructive feedback! It will significantly improve the clarity and robustness of our paper. Please let us know if you have further questions and concerns!

---

> > ### Comment · Reviewer_4Zzx · 2025-06-09
> > **Reply to Authors**
> >
> > Thanks for authors' response. They almost resolve my concerns, and I don't have further questions. I will keep my score as marginally above acceptance.

---

> ### Comment · Area_Chair_xRGV · 2025-06-05
>
> Hi Reviewer 4Zzx! Just a reminder that the discussion period for COLM papers has begun. Could you please take a look at the author response to review and let them know whether it addresses any of your outstanding questions?
>
> Thanks,
> Your AC

---

### Official Review · Reviewer_5EYn · 2025-05-12

**Rating:** 7
**Confidence:** 3
**Ethics Flag:** 1

**Summary:**

The paper proposes UQAC, a new method for uncertainty quantification in LLMs, which uses attention chains to identify important tokens in the reasoning process, and calculates the marginalization of these selected tokens as uncertainty.

**Questions To Authors:**

EXTRA REFS:

[1] AoE: Angle-Optimized Embeddings for Semantic Textual Similarity (ACL 2024)
[2] Fantastic Semantics and Where to Find Them: Investigating Which Layers of Generative LLMs Reflect Lexical Semantics (ACL 2024）

TYPOS:

- Line 17: reduces --> reduce
- Line 44: Further --> Furthermore

SUGGESTIONS

- Given reproducibility, the authors should include the prompt, and settings of parameters such as temperature and top_p in the experiments setting within the paper.

-Figure 4 shows that there is a significant difference in the reasoning sequence lengths between correct and incorrect answers on the MATH dataset, which may have influenced the results. Authors can consider conducting a controlled
variable experiment to analyze the impact of sequence length on the UQ methods.

- The experimental results (Table 1) show that UQAC exhibits significant performance differences across different datasets and models. Additionally, different variants of UQAC also show considerable variation in performance on the same dataset. Authors can try to include more analysis to explain the reasons behind these differences.

**Reasons To Accept:**

This paper uses attention chains to filter the crucial tokens in CoT, which is an innovative approach in UQ. This method doesn't need finetuning or multiple times inference, which is computationally efficient.
The entire paper framework is well-founded and logically coherent, especially for the extraction of the attention chain and the utilization of it (attention approximation, similarity approximation, answer approximation).
The experiments in the paper are comprehensive: the authors use multiple reasoning datasets and compare UQAC with numerous baselines using multiple evaluation metrics, such as AUROC and ECE.

**Reasons To Reject:**

In the Similarity-Based Filtering, the cosine similarity metric relies solely on the model’s hidden states and suffers from saturation zones that lead to vanishing gradients, yet the paper does not compare it against other effective similarity measures, such as[1]. Also,
recent studies find that the last hidden layer may not the best choice for semantic information extraction, such as[2].

In line 254, the setting of several hyperparameters, value lacks detailed evidence or ablation studies.

From the experimental results (Table 1), it can be seen that Self-Consistency remains the best-performing method overall, while some variants of the proposed UQAC method show better performance than Self- Consistency in very few cases.

---

> ### Author Response · Authors · 2025-05-31
> **Response to Reviewer 5EYn**
>
> We sincerely thank Reviewer 5EYn for the detailed review and constructive recommendations. Below, we address each point accordingly and describe the updates we will make to our manuscript:
>
> - **Typos**: We appreciate the careful review and have corrected the indicated errors.
>
> - **Semantic Similarity Issues**: We thank you for introducing these relevant works, which highlight potential improvements in semantic similarity evaluation. While we did not originally incorporate alternative similarity measures due to the scope of the initial development, we acknowledge the validity of your suggestion. We plan to explore these advanced semantic similarity methods in future iterations to further enhance the robustness of UQAC.
>
> - **Experiment Setup Details**: To enhance reproducibility, we will add the detailed experimental settings in the updated manuscript, specifically:
>
>   - Sampling temperature is set to 0 for all baselines except self-consistency, which uses a temperature of 0.5;
>   - The parameter top\_p is consistently set to 1;
>   = Prompting schemes are briefly discussed in Lines 616--624. Specifically, on MATH and GSM8k, all models receive only the question as input, except for gemma-2, which includes the prompt "\n\nPut your final answer in `\boxed{ }`." For BBH, we use three provided in-context examples with answers enclosed in `\boxed{ }`. Further implementation details are available in our code repository. We'll include a detailed discussion on this topic in the revised manuscript
>
> - **Controlled Experiment on Reasoning Length**: We appreciate this insightful suggestion. However, controlling reasoning sequence length without affecting the model’s inherent reasoning capabilities poses certain challenges. We have a short discussion on this topic in Lines 264--268 and appendix D.2 and remain open to suggestions on better experimental designs to address this variable.
>
> - **Further Analysis on Experimental Variations**: We recognize the significant variation in performance across datasets and UQAC variants. We attribute differences across datasets primarily to inherent model calibration on the specific datasets. Notably, all three theoretically distinct methods—Self-Consistency, Verbalized Uncertainty, and UQAC—demonstrate notably lower performance on GSM8k in terms of AUROC. We discuss this phenomenon briefly in section D.1. Regarding variations between UQAC methods, the performance difference arises from approximations in marginalizing joint reasoning probabilities and answer probabilities, introducing complexities and noise. Ultimately, the marginal approximation (\$\tilde{P}\$) is more theoretically robust and consistently performs better in terms of stability across AUROC and ECE, as detailed in Lines 290--296.
>
> - **Hyper-Parameters**: We conducted additional hyper-parameter studies and will include the results and analysis in the revised version. The results are in the Tables below. In short, It is possible to see (much) better scores with hyper-parameters tuned for models and datasets, but the parameter set we provide in the paper is generally stable across different setups.
>
> Your feedback significantly improves our manuscript, and we thank you again for the thoughtful review. Please let us know about your thoughts!
>
> Table 1. Hyper-parameter study of Llama 3.2 1B on GSM8k. The first column is organized as < n attention heads $K$ >-< attention weight $\theta$>-< similarity filtering target length $L'_{attn}$ >-< similarity filtering threshold >
> |                 | AUC             | ECE             |
> |:----------------|:----------------|:----------------|
> | 8-0.3-10-0.0    | 0.6538 ± 0.0105 | 0.1808 ± 0.0075 |
> | 8-0.5-10-0.0    | 0.6509 ± 0.0112 | 0.1862 ± 0.0098 |
> | 8-0.7-10-0.0    | 0.6561 ± 0.0108 | 0.1764 ± 0.0082 |
> | 16-0.3-10-0.0   | 0.6442 ± 0.0121 | 0.1871 ± 0.0107 |
> | 16-0.5-10-0.0_x | 0.6415 ± 0.0112 | 0.1888 ± 0.0102 |
> | 16-0.7-10-0.0   | 0.6462 ± 0.0138 | 0.1910 ± 0.0111 |
> | 32-0.3-10-0.0   | 0.6388 ± 0.0145 | 0.1952 ± 0.0137 |
> | 32-0.5-10-0.0   | 0.6376 ± 0.0133 | 0.1924 ± 0.0124 |
> | 32-0.7-10-0.0   | 0.6330 ± 0.0145 | 0.1932 ± 0.0138 |
> | 16-0.5-8-0.0    | 0.6514 ± 0.0104 | 0.2385 ± 0.0097 |
> | 16-0.5-8-0.2    | 0.6481 ± 0.0100 | 0.2410 ± 0.0096 |
> | 16-0.5-8-0.5    | 0.6190 ± 0.0101 | 0.3133 ± 0.0091 |
> | 16-0.5-10-0.0_y | 0.6475 ± 0.0119 | 0.1954 ± 0.0127 |
> | 16-0.5-10-0.2   | 0.6421 ± 0.0122 | 0.2002 ± 0.0130 |
> | 16-0.5-10-0.5   | 0.6168 ± 0.0113 | 0.2947 ± 0.0116 |
> | 16-0.5-12-0.0   | 0.6360 ± 0.0112 | 0.1967 ± 0.0095 |
> | 16-0.5-12-0.2   | 0.6301 ± 0.0114 | 0.2012 ± 0.0091 |
> | 16-0.5-12-0.5   | 0.6155 ± 0.0108 | 0.2932 ± 0.0101 |

---

> > ### Author Response · Authors · 2025-05-31
> > **Tables (cont.)**
> >
> > Table 2. Gemma-2-2b on GSM8k
> > |                 | AUC             | ECE             |
> > |:----------------|:----------------|:----------------|
> > | 8-0.3-10-0.0    | 0.6836 ± 0.0068 | 0.3577 ± 0.0043 |
> > | 8-0.5-10-0.0    | 0.6843 ± 0.0061 | 0.3572 ± 0.0040 |
> > | 8-0.7-10-0.0    | 0.6880 ± 0.0052 | 0.3594 ± 0.0033 |
> > | 16-0.3-10-0.0   | 0.6830 ± 0.0106 | 0.3528 ± 0.0039 |
> > | 16-0.5-10-0.0_x | 0.6810 ± 0.0099 | 0.3545 ± 0.0047 |
> > | 16-0.7-10-0.0   | 0.6799 ± 0.0093 | 0.3568 ± 0.0034 |
> > | 32-0.3-10-0.0   | 0.6924 ± 0.0099 | 0.3470 ± 0.0026 |
> > | 32-0.5-10-0.0   | 0.6899 ± 0.0093 | 0.3492 ± 0.0021 |
> > | 32-0.7-10-0.0   | 0.6919 ± 0.0075 | 0.3505 ± 0.0011 |
> > | 16-0.5-8-0.0    | 0.6914 ± 0.0077 | 0.3897 ± 0.0035 |
> > | 16-0.5-8-0.2    | 0.6918 ± 0.0076 | 0.3899 ± 0.0036 |
> > | 16-0.5-8-0.5    | 0.6792 ± 0.0068 | 0.4032 ± 0.0036 |
> > | 16-0.5-10-0.0_y | 0.6810 ± 0.0099 | 0.3545 ± 0.0047 |
> > | 16-0.5-10-0.2   | 0.6814 ± 0.0097 | 0.3547 ± 0.0049 |
> > | 16-0.5-10-0.5   | 0.6670 ± 0.0072 | 0.3778 ± 0.0032 |
> > | 16-0.5-12-0.0   | 0.6789 ± 0.0093 | 0.3059 ± 0.0044 |
> > | 16-0.5-12-0.2   | 0.6794 ± 0.0091 | 0.3060 ± 0.0047 |
> > | 16-0.5-12-0.5   | 0.6643 ± 0.0063 | 0.3398 ± 0.0026 |
> >
> > Table 3. Qwen-2.5-1.5B on GSM8k
> > |                 | AUC             | ECE             |
> > |:----------------|:----------------|:----------------|
> > | 8-0.3-10-0.0    | 0.5830 ± 0.0070 | 0.3490 ± 0.0034 |
> > | 8-0.5-10-0.0    | 0.5868 ± 0.0087 | 0.3485 ± 0.0022 |
> > | 8-0.7-10-0.0    | 0.5928 ± 0.0051 | 0.3462 ± 0.0023 |
> > | 16-0.3-10-0.0   | 0.5773 ± 0.0055 | 0.3498 ± 0.0030 |
> > | 16-0.5-10-0.0_x | 0.5753 ± 0.0060 | 0.3511 ± 0.0028 |
> > | 16-0.7-10-0.0   | 0.5827 ± 0.0041 | 0.3494 ± 0.0020 |
> > | 32-0.3-10-0.0   | 0.5816 ± 0.0101 | 0.3538 ± 0.0005 |
> > | 32-0.5-10-0.0   | 0.5862 ± 0.0098 | 0.3569 ± 0.0025 |
> > | 32-0.7-10-0.0   | 0.5857 ± 0.0082 | 0.3499 ± 0.0043 |
> > | 16-0.5-8-0.0    | 0.6030 ± 0.0040 | 0.3886 ± 0.0037 |
> > | 16-0.5-8-0.2    | 0.6066 ± 0.0049 | 0.3893 ± 0.0034 |
> > | 16-0.5-8-0.5    | 0.4807 ± 0.0064 | 0.4987 ± 0.0085 |
> > | 16-0.5-10-0.0_y | 0.5797 ± 0.0060 | 0.3476 ± 0.0027 |
> > | 16-0.5-10-0.2   | 0.5817 ± 0.0073 | 0.3520 ± 0.0032 |
> > | 16-0.5-10-0.5   | 0.4775 ± 0.0055 | 0.4787 ± 0.0066 |
> > | 16-0.5-12-0.0   | 0.5813 ± 0.0071 | 0.3174 ± 0.0056 |
> > | 16-0.5-12-0.2   | 0.5864 ± 0.0079 | 0.3205 ± 0.0064 |
> > | 16-0.5-12-0.5   | 0.4765 ± 0.0057 | 0.4608 ± 0.0064 |
> >
> > Table 4. Gemma-2-2B on MATH
> > |                 | AUC             | ECE             |
> > |:----------------|:----------------|:----------------|
> > | 8-0.3-10-0.0    | 0.7331 ± 0.0064 | 0.1577 ± 0.0071 |
> > | 8-0.5-10-0.0    | 0.7356 ± 0.0067 | 0.1501 ± 0.0083 |
> > | 8-0.7-10-0.0    | 0.7387 ± 0.0073 | 0.1419 ± 0.0082 |
> > | 16-0.3-10-0.0   | 0.7392 ± 0.0107 | 0.1568 ± 0.0056 |
> > | 16-0.5-10-0.0_x | 0.7408 ± 0.0119 | 0.1523 ± 0.0073 |
> > | 16-0.7-10-0.0   | 0.7397 ± 0.0116 | 0.1476 ± 0.0095 |
> > | 32-0.3-10-0.0   | 0.7422 ± 0.0090 | 0.1462 ± 0.0064 |
> > | 32-0.5-10-0.0   | 0.7441 ± 0.0084 | 0.1459 ± 0.0045 |
> > | 32-0.7-10-0.0   | 0.7402 ± 0.0085 | 0.1386 ± 0.0044 |
> > | 16-0.5-8-0.0    | 0.7486 ± 0.0096 | 0.1725 ± 0.0041 |
> > | 16-0.5-10-0.0_y | 0.7409 ± 0.0118 | 0.1523 ± 0.0073 |
> > | 16-0.5-12-0.0   | 0.7418 ± 0.0116 | 0.1411 ± 0.0093 |
> >
> > Table 5. Gemma-2-2B on BBH
> > |                 | AUC             | ECE             |
> > |:----------------|:----------------|:----------------|
> > | 8-0.3-10-0.0    | 0.5760 ± 0.0101 | 0.2576 ± 0.0050 |
> > | 8-0.5-10-0.0    | 0.5811 ± 0.0061 | 0.2550 ± 0.0052 |
> > | 8-0.7-10-0.0    | 0.5932 ± 0.0079 | 0.2525 ± 0.0038 |
> > | 16-0.3-10-0.0   | 0.5643 ± 0.0081 | 0.2676 ± 0.0084 |
> > | 16-0.5-10-0.0_x | 0.5659 ± 0.0077 | 0.2639 ± 0.0096 |
> > | 16-0.7-10-0.0   | 0.5810 ± 0.0119 | 0.2561 ± 0.0082 |
> > | 32-0.3-10-0.0   | 0.5673 ± 0.0065 | 0.2664 ± 0.0065 |
> > | 32-0.5-10-0.0   | 0.5718 ± 0.0069 | 0.2601 ± 0.0065 |
> > | 32-0.7-10-0.0   | 0.5883 ± 0.0104 | 0.2545 ± 0.0068 |
> > | 16-0.5-10-0.0_y | 0.5654 ± 0.0090 | 0.2641 ± 0.0094 |
> > | 16-0.5-12-0.0   | 0.5663 ± 0.0105 | 0.2587 ± 0.0097 |
> > | 16-0.5-16-0.0   | 0.5925 ± 0.0097 | 0.2416 ± 0.0079 |
> >
> > The results of other model/dataset combinations show similar trends. We believe the included tables are sufficient to justify our discussion above. Let us know if the full results are preferred!

---

> ### Comment · Area_Chair_xRGV · 2025-06-05
>
> Hi Reviewer 5EYn! Just a reminder that the discussion period for COLM papers has begun. Could you please take a look at the author response to review and let them know whether it addresses any of your outstanding questions?
>
> Thanks,
> Your AC

---

### Official Review · Reviewer_tYEK · 2025-05-13

**Rating:** 8
**Confidence:** 3
**Ethics Flag:** 1

**Summary:**

The paper tackles the problem of quantifying the uncertainty of LLM's responses. The paper uses the common general assumption that we can quantify the uncertainty of an LLM's response by using the LLM's probability of the output sequence it generates. The paper argues that this probability is intractible to compute since the search space would be too large (though the paper doesn't mention it explicitly, I assume the search space is the size of the vocabulary raised to a number of tokens in the output). The proposed method cuts down the search space of which vocab terms to consider when creating an uncertainty measure, using the uncertainty of the attention distribution to determine which attention heads should be considered and then the tokens at each attention heads by using cumulative attention weights.

The paper uses quantitative experiments to demonstrate that the propose methods outperforms baselines in terms of quantifying the uncertainty of LLM's responses.

**Questions To Authors:**

- I am about confused about the baselines 1 and 2. Based on the paper, I thought these methods would be intractable and to hard to compute directly? Can you provide more information on how these baselines were implemented. If they are intractible to fully compute, was the a simple approach used where at each answer token, we just keep the vocabulary words that have the highest probability and then just recompute a soft-max on just those tokens?


- When I click through to the anonymous github, most of the “requested files are not found”. Only args.py in utils is available, There is no need to include the link in the submission if the code is not ready. its better not to include the link then

- related work starts out strong with discussing the issues with using comulative token-wise probabilities or entropies
- It would be nice for the transition from related work to section 3 (problem definition) to be a bit smoother, its a stark change with no smooth transition, though I understand this writing style


- In figure 1, it would be nice to show the final quantify for uncertainty. Is it close to 0 in this example?


- What if a Chain of thought sequence is not provided? It seem like this method won't work, am I correct or misunderstanding? its ok if this method requires a CoT sequence, just make that clearer in the intro and/or abstract

- lines 121-124 are redundant since this was already made clear in the related work, on the other hand I guess its good to repeat it, but its unnecessary
- the paper assumes that taking the probability of the response is a good way to quantify uncertainty, this is a common assumption so thats ok but more justification for this idea would be appreciated.

- line 254, put a footnote here reminding reading what these specific hyper-parameters control so that the reader doesnt have to jump back to those equations and discussion about those equations

- It would be nice if the paper included some qualitative examples & analysis

(this is a mix of comments & questions, no need to respond to all of them)

**Reasons To Accept:**

Paper provides sufficeint motivation, is well written and clear to understand.
The paper makes a nice contribution to the important problem of quantifying uncertainty and calibrating probabilities of LLM's outputs.
The paper includes good thorough quantitative experiments and analysis

**Reasons To Reject:**

None really from my perspective. The paper articulates the problem, a proposed method, and experiments to demonstrate how the approach can improve model calibration. There might be better techniques or baselines to compare against but I'd rely on those who are more familiar with state-of-the-art methods for model calibration for LLMs.

The method relies on generating a chain of thought response and an open-source LLMs, neither issues I see as strong reasons to reject the paper.

---

> ### Author Response · Authors · 2025-05-31
> **Response to Reviewer tYEK**
>
> We sincerely thank Reviewer tYEK for the positive assessment and constructive feedback. Below, we address your specific questions and suggestions in detail:
>
> - **Clarification on Baselines 1 & 2**: Your understanding is correct. For baseline methods 1 and 2, we do not perform explicit marginalization over the entire vocabulary. Instead, we simplify by calculating the product of the probabilities for the actually sampled tokens, effectively sidestepping exhaustive marginalization. To further clarify the concern, we use `\bm{x}` as a sampled sequence (observation), and `\mathbf{x}` as a random variable, as stated in Table 3, category "Tokens & Sequences" and recommended by the conference.
>
> - **Code Accessibility**: We apologize for the confusion regarding the anonymous GitHub repository. It appears this is an issue on the anonymous GitHub. All files have remained intact and accessible. We recommend downloading the full repository directly via the "Download Repository" button in the top-right corner. This may provide access to all the code.
>
> - **Transition to and from Related Works**: We agree with your observation regarding the abrupt transitions. To enhance readability, we plan to reorganize our manuscript by relocating the Related Work section to follow the Experiments section, preceding the Conclusion. This restructuring should provide a smoother logical flow from Introduction to Method and facilitate easier comprehension.
>
> - **Figure 1 Visualization**: Thank you for suggesting improvements to Figure 1. Instead of altering this figure, we will include additional examples in the appendix to better illustrate the quantified uncertainty measures explicitly. For the example provided, the uncertainty is indeed low, as the computed confidence is high ($\tilde{P}=0.92$).
>
> - **Dependence on CoT**: Your interpretation is correct. UQAC explicitly requires the presence of a CoT reasoning sequence. When there is no reasoning chain, simpler methods such as baselines 2, 5, and even self consistency will become more applicable, lifting the necessity of applying UQAC.
>
> - **Redundancy in Lines 121--124**: Thank you for pointing out the redundancy. Given our planned reorganization of the related work section, we believe keeping this brief repetition in Section 3 maintains narrative clarity for readers who might skip directly to specific sections.
>
> - **Justification for Probability-Based Uncertainty**: We appreciate your suggestion regarding the justification for using probabilities as a measure of uncertainty. We will strengthen our manuscript by explicitly discussing prior empirical and theoretical justifications from existing literature to reinforce this common assumption.
>
> - **Hyper-Parameter Explanation**: Indeed, we initially included a longer experiment setup section with more detailed explanations, but we compromised it to meet the length limitation. Nonetheless, we'll do our best to provide a clearer description or at least include a comprehensive explanation in the appendix.
>
> - **Qualitative Examples & Analysis**: We acknowledge the value of providing qualitative examples. We plan to incorporate representative examples and accompanying analyses in the appendix of our revised manuscript, demonstrating concretely how UQAC improves calibration and provides intuitive uncertainty estimates.
>
> Thank you again for your detailed comments and questions. We believe addressing these points significantly improves our paper's clarity and comprehensibility. Please let us know if your questions have been adequately answered or if you have any additional ideas or suggestions!

---

> ### Comment · Area_Chair_xRGV · 2025-06-05
>
> Hi Reviewer tYEK! Just a reminder that the discussion period for COLM papers has begun. Could you please take a look at the author response to review and let them know whether it addresses your concerns?
>
> Thanks,
> Your AC

---

### Decision · Program_Chairs · 2025-07-08

**Decision:**

Accept

**Comment:**

For many applications we wish to estimate the *marginal* probability that an LM produces an output $y$ for an input $x$ following any reasoning chain $z$. But computing $p(y | x) = \sum p(y|z, x) p(z|x)$ is intractable if $z$s are very long strings, and Monte Carlo estimates tend to be high-variance because there are many ways to realize $z$ that don't ultimately affect a model's final answer. This paper presents a method that speeds up this calculation by using attention to identify "critical tokens" in the reasoning chain and approximating the marginal probability of generating those critical tokens, reducing the variability in $p(z|x)$ across different $z$ and ultimately the quality of the approximate $p(y|x)$. A majority of reviewers found the method well-motivated, the presentation clear, and the experimental results convincing. Reviewer ZMZs recommended against acceptance, on the basis that the interpretation of attention scores is itself contentious (e.g. https://arxiv.org/abs/1902.10186). But I agree with the authors that these concerns aren't really relevant in the context of this paper, as the particular use being made of attention here clearly *does* lead to improved results.

[Automatically added comment] At least one review was discounted during the decision process due to quality]